# Subcellular glycan-mannose receptor binding kinetics correlate with myeloid cell function

Kas Steuten [1], Johannes J. A. Bakker[1], Ward Doelman [1], Diana Torres-García[1], Amit Cherian[1], Christian Kurts [2], Roger Riera[3], Lorenzo Albertazzi [3] & Sander I. van Kasteren [1,4,5] ✉

Extracting single-molecule lectin binding kinetics from primary cells has not been possible to date. Here, we present Glyco-PAINT-APP (Automated Processing Pipeline), an automated method that enables the extraction of sub-cellular glycan interaction kinetics using a Points Accumulation for Imaging in Nano-Topography (PAINT)-based approach. This approach leverages an algorithm for precise, high-throughput, subcellular analysis of glycan binding dynamics and facilitates functional correlation studies between glycoform binding patterns and immune cell polarization. Using synthetic glycans and glycosylated antigens, we demonstrate the ability of the technique to automatically correlate glycan binding parameters in subregions of dendritic cell membranes with increased uptake and cross-presentation efficiency of these antigens. Additionally, we show how the method can uncover subtle differences in MR-mediated glycan binding across various MR-expressing primary cells and cell lines. Taken together, Glyco-PAINT-APP enables insights into the cell-intrinsic heterogeneity of glycan-structure-activity relationships in myeloid immune cells.

Myeloid immune cells, such as dendritic cells (DCs) and macrophages are key sentinels of the immune system. They sense the presence of microbial species or altered self through pattern recognition receptors (PRRs), which in turn can trigger innate and adaptive immune responses. One important family of PRRs are the carbohydrate-binding proteins of the C-type lectin receptor family (CLRs). CLR ligation can induce various immune functions, including endocytosis, polarization, cytokine secretion, cell adhesion, and motility, which can subsequently trigger downstream immune responses[1,2].

The signalling pathways leading to these biological outcomes are intricate and finely tuned, which complicates the study of how ligand recognition leads to signalling, particularly in their native environment on the cell surface. This complexity arises from a variety of factors: binding of the receptors to ligands is weak and often multivalent, multiple receptors have overlapping specificities, and the surface expression patterns of CLRs can be highly dynamic. Myeloid cells express a diverse array of CLRs for sensing aberrant glycosylation. These can exist as monomers, multimers, and even form hetero-dimeric complexes[3,4]. Various factors, such as the polarization state of the cell, can modulate the expression levels of these lectins and alter the sub-cellular distribution between the surface and intracellular storage pools[5,6]. The resulting dynamic and heterogeneous surface expression leads to significant variability in receptor-ligand interactions.

[1]Department of Chemical Biology and Immunology, Leiden Institute of Chemistry, Leiden University, Leiden, the Netherlands. [2]Institute of Molecular Medicine and Experimental Immunology, University Hospital of Bonn, Rheinische Friedrich Wilhelm University, Bonn, Germany. [3]Department of Biomedical Engineering and Institute for Complex Molecular Systems, Eindhoven University of Technology, Eindhoven, the Netherlands. [4]Institute of Chemical Immunology, Leiden University, Leiden, The Netherlands. [5]Institute of Chemical Neuroscience (iCNS), Leiden University, Leiden, The Netherlands. ✉e-mail: s.i.van.kasteren@chem.leidenuniv.nl

To complicate matters further, myeloid cells also present a repertoire of glycans that can serve as ligands for their own lectins. These can modulate signalling pathways through ligand competition and receptor clustering. Polarization of immune cells also affects the expression of these cis-ligands, adding yet another layer of complexity to carbohydrate-immune lectin biology[7,8]. The final complicating factor in CLR-binding and signalling is that the affinities of C-type lectin domains (CTLs) for their ligands are weak ($K_D$ in the μM to mM range)[9]. This complicates the detection and precise study of these interactions. Together with the intrinsic chemical complexity of glycan ligands, these factors have made it exceedingly difficult to establish clear relationships between glycan binding properties and lectin function, particularly as studying these lectins out of their native context removes many of these modulating factors.

We have recently reported a technique that took a step towards resolving some of these issues. We showed that carbohydrate-lectin binding kinetics could be quantified on the surface of lectin over-expressing cell lines (called Glyco-PAINT)[10]. Glyco-PAINT exploits the transient interaction between fluorescently-labelled sugars and their receptors to generate single molecule binding events[10], analogous to those induced by the mismatches of DNA-strands in conventional Points Accumulation for Imaging in Nanoscale Topography (DNA-PAINT)-imaging[11]. Accumulation of these binding events over time allows for the mapping of receptor-ligand interactions on the cell surface and subsequent derivatization of kinetic parameters such as on-rate, off-rate and diffusion coefficients on a per-cell basis derived from single-molecule binding information.

We envisaged that Glyco-PAINT could be a powerful technique to unravel the complexity of myeloid lectin binding, too, as the single molecule as well as the live cell aspects of the technique are ideal for studying these capricious receptors within their native environment on the myeloid cell surface. However, three major hurdles needed to be overcome to achieve this. The first is that Glyco-PAINT had only been demonstrated in an over-expression system. Second, the Glyco-PAINT processing pipeline could only be used to obtain the binding parameters averaged over the surface of a whole cell, which we hypothesized to be insufficient due to the microheterogeneity of receptor expression on primary cells. The final problem relates to the probes used in the experiments. Glyco-PAINT-probes consist of a carbohydrate cluster directly attached to a fluorophore. This precludes the evaluation of how the immunological outcomes change with receptor binding behaviour.

One receptor, which is archetypal in its biological complexity, is the mannose receptor (CD206, MR). This lectin is expressed on cells of the myeloid lineage[12–14], where it carries out a variety of functions. It acts as an endocytic receptor to clear soluble glycoproteins and as a phagocytic receptor of glycosylated particles, leading to clearance and antigen presentation[15–17]. It also acts as a mediator of inflammatory signalling by inducing M2 polarization of macrophages and inducing T cell tolerance[18,19]. How the MR translates ligand recognition to all these separate functions remains unknown. Structurally, the MR contains multiple functional domains, including an N-terminal cysteine-rich domain capable of binding sulphated glycans (that can inhibit receptor function[20]), a fibronectin-like domain capable of binding collagen peptides[21,22], and eight CTLs, of which only CTL4 and CTL5 exhibit functional calcium-dependent binding of neutral sugars[23]. These domains are further modulated by glycosylation at seven N-linked sites, influencing receptor binding properties[24]. Moreover, the intracellular tail lacks classical signalling motifs[5,15], though evidence suggests phosphorylation and diaromatic motifs may facilitate endosomal sorting and signalling[25–27].

Despite this biological complexity, the MR has been under heavy investigation as vaccine targeting receptor. Ligation of the MR was shown to lead to the enhanced cross-presentation (and thus cytotoxic CD8 T-cell activation) of glycoprotein antigens, which can be employed to enhance the anti-cancer and anti-viral properties of therapeutic vaccines[28]. However, further pursuit of this phenomenon has led to conflicting results[29–37]. No studies have yet managed to correlate the binding of specific ligands to the MR to downstream biology (i.e., uptake and/or cross-presentation). Studies using bulk methods, such as surface plasmon resonance (SPR), failed to correlate binding affinities with uptake or cross-presentation, highlighting the need for methodologies capable of quantifying receptor-ligand kinetics in the native cellular context[10,30].

Here, we present the redevelopment of Glyco-PAINT, such that we can obtain binding parameters of glycans to the MR on myeloid cells. We have developed an analysis algorithm to independently process large volumes of Glyco-PAINT recordings of glycan binding events to primary immune cells in such a way that binding event concentration, dwell times, diffusion coefficients and displacement can be determined for the highly heterogeneous and disperse binding observed across the cell basal membrane of DCs and macrophages. We have dubbed this method "Glyco-PAINT-Automated Processing Pipeline" (Glyco-PAINT-APP). In addition, we developed a with probes containing a pendant cross-presentable epitope between the glyco-clusters and fluorophore of the earlier incarnations of the probe. This allowed us to quantify lectin binding kinetics and correlate this with the uptake and cross-presentation of the same construct.

Correlating the binding kinetics to uptake and cross-presentation biology yielded some surprising findings. We found that the binding of the glyco-clusters was very heterogeneously distributed over the surface of the DCs. On a single cell, highly varying binding regions were observed: from areas with no probe binding, to regions with low numbers of binding events to a highly mobile MR, to unique sub-cellular population of areas where probes engaged with the cell surface in an MR-independent manner. We found a strong negative correlation (Pearson's R: −0.90) between these highly mobile receptors and probe endocytosis by the DC, which was highly MR-dependent. In contrast, cross-presentation of mannosylated antigens showed a strong positive correlation only to the residence time of the specific probe on DCs of both WT and MR$^{-/-}$ origin (Pearson's R: 0.82 and 0.77, respectively), independent of the amount of surface mobility. In a second application of the approach, we could also quantify changes in mannoside binding during the macrophage polarisation trajectory and found that the cumulative duration of binding events was enhanced for the M2 phenotype in accordance with upregulated MR expression and endocytosis of ligands.

## Results

### Quantifying glycan-lectin binding on primary DC

Our first aim was to determine whether the Glyco-PAINT methodology could be used to study binding events on the surface of primary bone-marrow derived mouse dendritic cells (BMDCs)[38]. Unlike the over-expression system used previously, BMDCs are characterized by heterogeneous and dynamic receptor expression of multiple lectins, multiple of which are capable of binding mannosylated ligands, and structural features such as tight focal adhesions and podosomes that may exclude binding events from the TIRF plane[39,40].

Preliminary attempts to quantify binding kinetics of a labelled mannose glycan library consisting of previously reported[11] glyco-clusters **8**, **10**, **11** and **13** on BMDCs yielded an unexpected result: whereas binding events could clearly be seen in the Glyco-PAINT recordings (Fig. 1b and Supplementary Fig. 1A–D) no significant differences were observed between the probes for either the events μm$^{-2}$s$^{-1}$ (a relative measure of $k_{on}$), nor in the average ligand dwell time ($τ$, which is the inverse off-rate, or $k_{off}^{-1}$) (Supplementary Fig. 1E, F). When comparing control experiments in CHO-MR cells did show the binding behaviour as originally reported (Fig. 1a and Supplementary Figs. 2A–D and 2E, F for quantification).

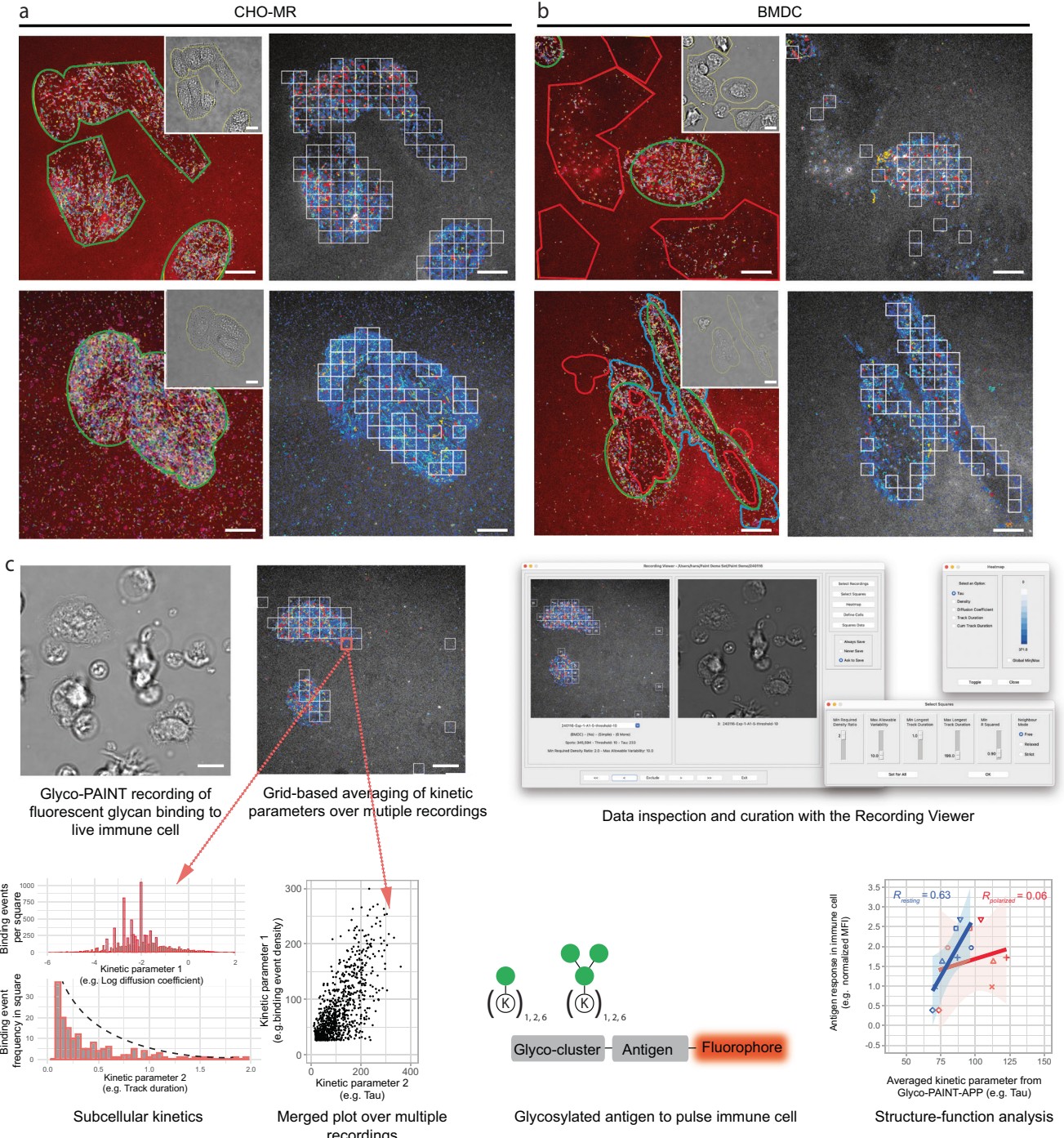

**Fig. 1 | Glyco-PAINT-APP allows for subcellular analysis of immune cell lectin binding. a** Left images: original Glyco-PAINT analysis of fluorescent mannose glycan binding to CHO-MR cells. ROIs are drawn using brightfield-defined cell contours (inset, yellow outlines), and areas where binding events are overlapping with cell areas are outlined in green. Binding trajectories are randomly coloured. Right: Grid-based identification of binding events using Glyco-PAINT-APP. Automatically defined white squares show an independent selection of cell-related binding events and define the area over which events are integrated. **b** Same as **a** but for BMDC. In the left images, manual outlining of non-overlapping areas devoid of binding events arising from primary cell intrinsic heterogeneity in red and areas engaging in binding events but outside of brightfield-defined cell outlines in blue. **c** Overview of the Glyco-PAINT-APP workflow: primary immune cells are cultured in a live-cell imaging chamber, and a brightfield image is recorded. After addition of a fluorescent probe, spots are recorded using PAINT imaging and reconstructed into tracks reflecting glycan binding events which are then subdivided into squares of variable size depicting subcellular regions of live immune cells. Kinetic parameters such as on-, off-rates and diffusion coefficients of glycan ligands averaged per individual square can then be interpreted, analysed and filtered using the *Recording Viewer* tool and displayed as heatmaps, histograms or scatterplots. Kinetic parameters derived using the algorithm can subsequently be correlated to functional effects of resting or polarized immune cells. Scalebars represent 10 μm, data figures represent illustrative data.

Interestingly, the BMDC image reconstructions showed a very high variance in the density of binding events (density maps) between individual cells—with some cells devoid of binding events, and other showing large numbers of events—and even between different regions of the basal membrane of a single cell (Fig. 1b). Areas of high binding events were observed overlapping with brightfield-defined cell outlines (Fig. 1b green areas and Supplementary Fig. 1A–D). However, large areas of the cell basal membrane were also devoid of any binding event density (Fig. 1b red areas and Supplementary Fig. 1A–D), and some areas outside the brightfield-defined cell surface appeared to be also engaging in binding events (Fig. 1b, blue areas), suggesting the presence of receptors on the ultrathin dendrites. This heterogeneity was not observed for CHO-MR cells (Fig. 1a, green areas and Supplementary Fig. 2).

These observations meant that the fundamental assumption underpinning the Glyco-PAINT method—that the receptor density [R] is homogeneous over the whole basal membrane of every cell— was incorrect for BMDCs. This in turn implicates that the per-cell relative on-rate was an irrelevant parameter for the study of primary cells. The off-rates, too, are averaged on a per-cell basis as they are integrated over the complete area; only a subsection contains binding-enabled lectins. Together, the observed highly heterogeneous binding event distribution was calling for a different surface area definition to obtain meaningful quantification of kinetic parameters.

## Glyco-PAINT-APP

To address the heterogeneity observed on BMDC surfaces, we developed the Glyco-PAINT-Automated Processing Pipeline (Glyco-PAINT-APP), where instead of using the whole cell as the averaging unit, the basal membrane was segmented into squares to account for the subcellular variation of binding events. This allows for a more precise quantification of glycan binding kinetics within localized areas whilst ignoring the areas in which no binding events occur. An additional boon to this approach was that this method enabled automation of the workflow, as compared to the previous requirement for a manual identification of the cell outlines, thereby accelerating analysis speed. The Glyco-PAINT-APP is schematically outlined in Fig. 1c (and described in full detail in the Supplementary Software Manual in the Supplementary Information).

Glyco-PAINT-APP starts by subjecting sets of microscope recordings of Glyco-PAINT experiments to single particle tracking analysis (see also Supplementary Video 1). This yields output files containing spatial information belonging to each spot in each frame of the Glyco-PAINT recording. Tracks, consisting of closely linked spots through multiple consecutive frames, can then be identified and defined by their duration, and movement-based parameters (such as distance travelled, start-point, end-point, and displacement). The Glyco-PAINT-APP then processes the recordings by segmenting the field of view in a raster of squares (typically 20 × 20 squares per field of view), rather than manually outlining cells. The kinetic information is then quantified for each of these squares, rather than per cell. This allowed us to select only those regions of the cell surface on which binding events take place based on the number of events that occur for a given set of quality control parameters. Squares are selected based on the quality control parameters derived from the TrackMate output. In addition, we have included new quality control measures, namely the "*Density Ratio*" (the ratio of events in a square divided by the number of events in the lowest binding 10% of squares that contain at least one event), and the "*Maximum Variability*" (a score for event homogeneity within a single square). In addition, "*Connectivity*" (a measure for how isolated a square can be within the image) can also be included but is set to "free" in these experiments. Further selection takes place via the "*R Squared*" value that relates to the quality ($R^2$) of the one-phase exponential decay fit or "*Minimum number of Tracks To Calculate Tau*" of (see Supplementary Software Manual in the Supplementary Information for a detailed description of quality control parameters). The interactive

tuning of these parameters is done using the "*Select Squares*" window and the effects on the recording are assessed by direct visual feedback from the "*Recording Viewer*" (Fig. 1c and Supplementary Video 2), where the tracking overlay with grids and brightfield image are displayed side-by-side. This allows for user-informed data curation.

Once the squares are selected, kinetic parameters including glycan on- and off-rates, diffusion coefficients, speed, displacement and maximum, average, or total binding event duration can be displayed for each square per recording using the heatmap toggle function in the "*Recording Viewer*" (Fig. 1c). The "*Compile Project*" built-in modality can integrate kinetic data from multiple recordings obtained from multiple biological replicates within a single project and can generate a merged output file to which statistical analysis can be applied with scientific graphing software of choice (R Studio[41], GraphPad Prism, etc.).

With this automated analysis pipeline in place, we first validated the approach by analysis of the binding of glycoclusters **8, 10, 11** and **13**. These ligands cover a range of affinities from our earlier Glyco-PAINT work and helped determine how variation of each parameter affected $\tau$ (and thus off-rate), track mobility, the relative on-rate and density ratio. This sweep included varying spot detection thresholds, track reconstruction parameters, and grid size. The parameter sensitivity analysis for tracking and spot detection parameters is depicted with heatmaps of each scenario in Supplementary Fig. 5A, B. From this analysis, we first determined how varying the parameters affected the number of squares that passed selection criteria per image for CHO-MR versus the MR-negative CHO-WT (Supplementary Fig. 6A). We found that any of the parameter values for Gap Closing Max Distance, Max Frame Gap and Linking Max Distance were able to select squares engaging in MR-mediated binding events from CHO-WT background. For the Minimum Nr Of Spots, we found that tracks should consist of a minimum of 3 spots, since the number of selected squares when allowing minimum track lengths of 2 spots in CHO-WT were boasted close to CHO-MR levels. This implicates that binding events spanning only 2 frames (100 ms) can be considered MR-unrelated background, whereas a minimum of 4 spots appeared to result in too stringent filtering (Supplementary Fig. 6A). For the Threshold, we found that values > 4 were resulting in accurate separation between the two cell types.

Next, we further analysed the sensitivity of track and spot detection parameters on the kinetic parameters off-rate, on-rate and diffusion coefficient. Here, it was found that of the tracking parameters, Min Nr Of Spots had a large influence on the average $\tau$ since short duration tracks are discarded. In contrast, median off-rates were largely independent of threshold settings above 3 (Supplementary Fig. 6B), median number of detected events (on-rate) remained constant above 5 (Supplementary Fig. 6C) and also diffusion coefficients did not vary above 4 (Supplementary Fig. 6D). However, the density ratio appeared to remain constant above a threshold of 8 (Supplementary Fig. 6E). To summarize, the track reconstruction parameters as indicated in the method section were kept constant for all analysis and the spot detection threshold was initially set at 5 since for this value the detection bias was minimized. If >1,500,000 spots per recording were detected, the threshold was increased to a maximum of 20 in steps of 5 due to computational limitations.

## Subcellular analysis of immune cell lectin binding

We next applied Glyco-PAINT-APP with these settings to the profiling of glycan binding events on DCs. For these intrinsically heterogeneous cells, we optimized the grid size to avoid averaging artefacts. For this we used the complete dataset of mannose ligands binding to BMDC, which consists of approximately 400 different recordings (each of standard 82 × 82 μm², 512 × 512 pixels, and 2000 frames at 20 fps). Our analysis demonstrated that 20 squares per axis provided an optimal balance between statistical robustness of the off-rate fit ($R^2 > 0.9$), the number of binding events per square, and the minimization of averaging artifacts (Supplementary Fig. 7A). Off-rates ($\tau$) did not vary with

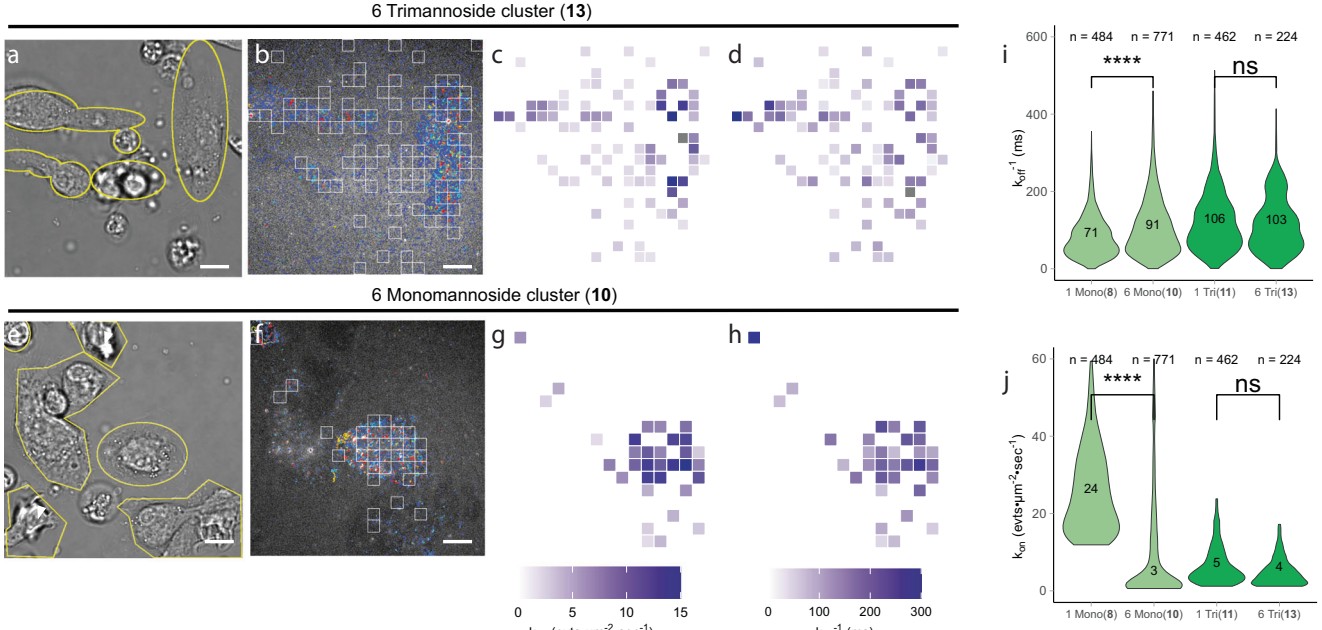

**Fig. 2 | Subcellular analysis of glycan binding on dendritic cells resolves averaging artefacts. a** brightfield image of live BMDCs. **b** Reconstruction of binding events recorded with 5 nM trimannoside glycan 13 processed using a 20 × 20 grid with all squares that have passed selection criteria (Density_Ratio > 2, R_Squared ≥ 0.9, Nr_Tracks/square > 20) displayed by white outlines (Supplementary Fig. 3 A, B show the same set of images for glycan clusters 11 and 13). **c** Heatmap projection of the number of binding events/square showing the high heterogeneity of binding

events even between the selected squares on the same cell. **d** Heatmap display of $k_{off}^{-1}$ per square. **e**–**h** The same as **a**–**d** but for glycan cluster 10. **i, j** Violin distribution plots and median value of $k_{on}$ and $k_{off}^{-1}$ for all glycans 10–13. $n$, depicts the number of squares that were analyzed per violin. Significance was assessed using two-way ANOVA followed by a Tukey post-hoc test. Scalebars represent 10 μm. Tracks in **b** and **f** are colored from short to long track length (light blue to red).

square size (Supplementary Fig. 7B), but the density of event (rel. $k_{on}$)-values, and the density ratio diverged with square size (Supplementary Fig. 7C, D).

With this analysis pipeline in hand, we set out to reprocess the recordings of the mannose probe library on primary DCs for which the original Glyco-PAINT procedure had not yielded any significant differences in binding between the different glycoforms. The square-based analysis for DCs is displayed in Fig. 2 (for the same recordings as in Fig 1a, Supplementary Fig. 1A–D, and extended in Supplementary Fig. 3A, B for mono- and trimannosides 8, 10, 11 and 13). Figure 2b, f show that the APP-algorithm was able to independently distinguish areas with binding event density from background using filtering based on the previously optimized parameter settings. The added advantage of this approach is that binding events that fell outside brightfield-invisible features of the DC are now also included in the analysis. The thus observed binding rates in different areas of the cell basal membrane can be visualised as heatmaps in Fig. 2c, d, g, h. The grid-based analysis by Glyco-PAINT-APP also improved the effective selection of cell areas engaged in binding events for mono- and trimannoside-based probe binding to CHO-MR (Supplementary Fig. 4), highlighting the omnipresence of microheterogeneity even in over-expression systems. Taken together, this method allows for precise analysis of glycan-lectin interactions on live immune cells with intrinsically heterogenic expression, avoiding the emergence of averaging artefacts as in the original Glyco-PAINT analysis. Some interesting parameters began to emerge from the approach. For example, a discrepancy between areas of high number of binding events (i.e., high rel.$k_{on}$; Fig. 2c, g) and areas with long average duration of binding (τ, $k_{off}^{-1}$, Fig. 2d, h) within the same cell could be seen on DCs.

## Uptake and antigen presentation of SLP glycoforms
We next wanted to determine whether any of the on-cell kinetic parameters could prove predictive for some of the previously reported

roles (see introduction) of the MR and related CLRs on BMDCs, namely its capacity as an uptake receptor and a cross-presentation enhancing receptor. We therefore designed Synthetic Long Peptide (SLP) versions of the fluorophore-labelled mannose clusters that also contained the model cross-presentable epitope Ovalbumin$_{247-264}$ (OVA SLP)[42], as it has been hypothesized that different glycoforms of these SLPs show different cross-presentation behaviour[33,43]. This additional feature would allow us to monitor the various binding parameters (by Glyco-PAINT-APP), uptake (by flow cytometry), and antigen cross-presentation (using the cognate OT-I T-cell), thereby having readouts for these three aspects of MR biology from a single probe.

The constructs were synthesized using in-line solid-phase peptide synthesis of the peptide antigen and an oligo-azidolysine (6-azido-norleucine) cluster, followed by on-resin modification with fluorophore. Copper-catalysed azide-alkyne cycloaddition (CuAAC)-mediated[44,45] coupling of the mannose glycans to the azidolysine clusters in solution yielded the target glycosylated SLPs. This resulted in mono-, bi- and hexavalent versions of mono- (**1-3**) or trimannosylated (**4-6**) SLPs and a non-glycosylated control molecule (**7**) that are displayed in Fig. 3a (synthetic procedures and characterization can be found in the supplementary information and Supplementary Fig. 8). These glycan motifs span a $K_D$ range for MR-binding from 3 μM to more than 100 μM (based on SPR studies[10,30]), thereby enabling investigation of glycan-structure-binding-activity relationships over a large affinity range.

To verify the absence of artefacts resulting from the SLP antigenic cargo on kinetics, we compared the median on- and off-rates of OVA SLP **1-6** to glycoclusters **8-13** on CHO-MR cells and plotted the correlation between the two in Fig. 3b, c. Neither extension of the peptide core nor the change of the fluorophore from ATTO655 to sulfo-Cy5 significantly affected $k_{off}$. A decrease in the median number of binding events per square was detected for the glycan SLPs with respect to the glycans **8-13**, potentially originating from modestly altered physical-chemical properties of the peptide cargo.

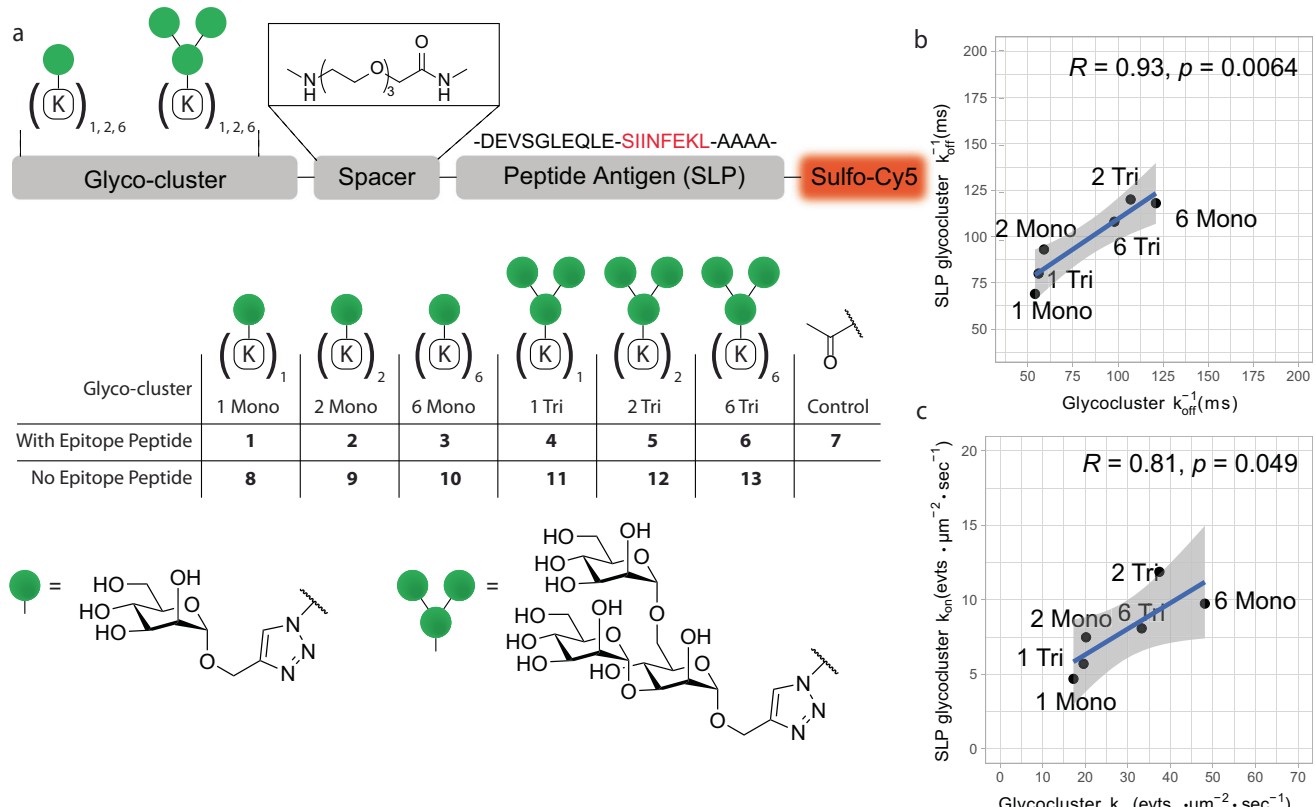

**Fig. 3 | Fluorescently labelled mannose ligands with SLP antigenic peptide retain MR-binding. a** Design of synthetic glycoclusters. 1–7 carry an N-terminal long peptide from Ovalbumin263-275 and a sulfo-Cy5 fluorophore on a C-terminal lysine side chain. Glycocluster series 8–13 do not contain this SLP but are directly attached to an ATTO655 fluorophore. **b**, **c** Correlation analysis of the median rel. $k_{on}$ and $k_{off}^{-1}$ between probes with identical glycan but bearing long peptide antigens binding to CHO-MR cells. Data points are medians for all squares analysed with a $20 \times 20$ grid and filtered with Density_Ratio > 2, R_Squared $\geq$ 0.9 and Nr_Tracks/Square > 20. Pearson's R and two-tailed $t$ test P value are displayed in the correlation plots, and the shaded area represents the 95% confidence interval of the regression line.

The binding parameters of these glycosylated SLPs were assessed for their binding to live WT BMDCs and BMDCs derived from MR$^{-/-}$ mice This analysis enabled us to precisely determine the contribution of glycan SLP binding events mediated by the MR, and not by any of the other lectins present on the DCs. A population of squares engaging in binding event with high mobility (high diffusion coefficient) and low event density ($k_{on}$) were uniquely present for the WT cells that were not observed on DCs lacking MR (Fig. 4a, b). This population of binding events appeared most prominent for glycan SLPs **2, 3** and **6** but not for **1, 5** and **7**. To ensure that uptake (and resulting removal of a ligand from the TIRF-plane) affected track length, cells were also treated with the uptake inhibitor Cytochalasin D (CytD, which works through inhibition of actin polymerisation)[46,47]. Treatment of cells with this inhibitor confirmed that uptake resulting from most ligand binding was reliant on endocytic mechanisms involving actin polymerisation, as event density, dwell time and diffusion coefficient increased significantly for probes 1 Tri (**4**) and 2 Tri (**5**) after CytD treatment of BMDC (Supplementary Fig. 9 for probe pairwise comparisons, Supplementary Fig. 14A, B for comparisons between WT, MR$^{-/-}$ and CytD conditions, and in Supplementary Table 1 numeric values are listed). These probes were also the most effectively taken up by DCs (vide infra).

Next, the uptake of these SLPs was measured by incubating BMDCs with SLP **1-7** for 1 h, followed by quantification of the sCy5 signal by flow cytometry. The histograms in Fig. 4c show that a specific subpopulation of MR$^+$CD11c$^+$ DCs (Supplementary Fig. 10B, E, for gating strategy and MR co-staining) are mostly engaged in this MR-mediated intracellular uptake (Supplementary Fig. 10D, for

competition with broad- spectrum ligand mannan and Supplementary Fig. 11A, B for intracellular localization of SLP **5** and **6**), highlighting the intrinsic heterogeneity of these cells. Quantification of this effect using phagocytic index (PI)[48] revealed a glycoform-specific profile with divalent trisaccharide-modified SLP **5** being most effectively taken up in contrast to its hexavalent counterpart **6**. Uptake of the non-glycosylated control **7** was near baseline levels, emphasizing the glycan-mediated engagement of endocytic lectins. Similarly, all uptake was abolished MR$^{-/-}$ cells (Supplementary Fig. 10B).

The cross-presentation efficiency of these glycosylated SLPs was tested next by pulsing differentiated DCs, which were matured for 2 h with TLR4 ligand MPLA, with 40 nM SLP (which is in a similar concentration range as Glyco-PAINT binding studies, see Supplementary Fig. 10C, D for different antigen dosages and alternative quantification of T cell proliferation) for 2 h followed by a 3-day coculture with cognate CD8 T cells that were freshly isolated from OT-I mice. The non-mannosylated control SLP **7** was most efficiently cross-presented, whereas, SLPs decorated with mannosides of increasing complexity and valency **1-6** showed an unexpected decrease in T-cell proliferation induction. We also evaluated the cross-presentation efficiency in MR$^{-/-}$ cells and found an identical glycoform-dependent pattern. To validate correct differentiation and functional response of the MR$^{-/-}$ cells as compared to WT cells we characterized the change in cell surface marker expression levels in response to LPS stimulus and found no aberrant expression patterns Supplementary Fig. 10A. Additionally, we verified DC origin (see Supplementary Fig. 11F for steady-state cross-presentation of antigens in splenic DCs[33]) and the effect of timing of DC maturation and found an

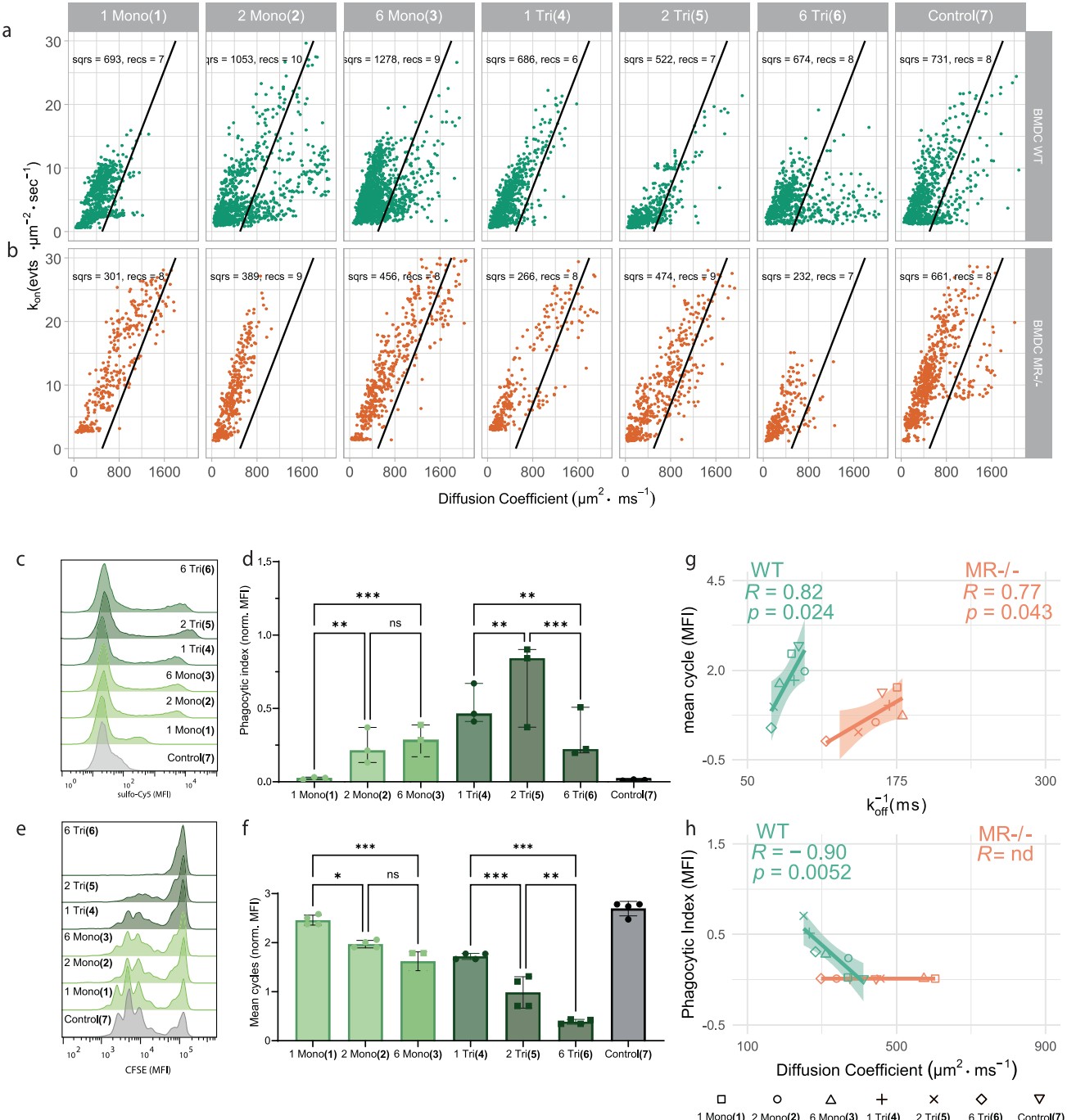

**Fig. 4 | Unique population of MR-specific binding events and kinetic parameters correlate with ligand functionality. a** Scatterplots of the rel. $k_{off}$ and diffusion coefficient per square for binding events between probes 1-7 and WT BMDC. Number of accepted squares and recordings are written in the plots; the diagonal black line represents a manually drawn gate that identifies MR-mediated binding events in WT cells. **b** Same as in **a** for MR$^{-/-}$ BMDC. **c** Histograms of BMDCs incubated with probe 1-7 for 1 h and subsequent measurement of endocytosis in CD11c+ cells using flow cytometry. **d** Bar chart of **c** as the phagocytic index. **e** Histograms of CFSE dilution in dividing OT-I T cells that were cocultured for 3 days with mature BMDC that were pulsed for 2 h with antigens 1-7 and measured

using flow cytometry. **f** Bar chart of the average number of cell divisions per T cell obtained from (**e**). **g** Correlation between median $k_{off}^{-1}$ for WT or MR$^{-/-}$ BMDCs and the mean number of T-cell divisions. **h** Correlation between probe median diffusion coefficients for WT or MR$^{-/-}$ cells and phagocytic index. Glyco-PAINT-APP analysis was done using a 20 × 20 grid and filtered with Density_Ratio > 2, R_Squared ≥ 0.9 and Nr_Tracks/Square > 20. Significance was assessed using two-way ANOVA followed by a Tukey post-hoc test. Pearson's R and two-tailed $t$-test P value are displayed in the correlation plots, and the shaded area represents the 95% confidence interval of the regression line. Datapoints in **d** and **f** represent 3 and 4 biological replicates, respectively, bars and error bars represent mean ± interquartile range.

---

identical pattern for all timepoints in agreement with the observed diminishing effect of antigen mannosylation on T cell proliferation (Supplementary Fig. 11E, G).

Having observed this non-linear (Supplementary Fig. 12A) and, to us at least, counter-intuitive relationship between binding, the

amounts of endocytosed antigen, and MHC-I-restricted cross-presentation, we next determined whether any of the subcellular kinetic parameters of SLP binding to DCs were predictive of either of these biological functions. A correlation was made using the median values of the following 10 kinetic parameters as derived by Glyco-PAINT-APP

per analysed square: $\tau$ (ms), rel. $k_{on}$ (events s$^{-1}$μm$^{-2}$), total track duration (s), long track duration (10% longest, s), short track duration (90% shortest, s), diffusion coefficient (nm$^2$s$^{-1}$), speed (μm s$^{-1}$) and max. speed (μm s$^{-1}$). From this analysis, glycan-SLP residence time ($\tau$) on the DC surface was the best predictor for cross-presentation and this correlation upheld for the MR$^{-/-}$ cells. Furthermore, high diffusion coefficients (and accordingly larger displacement of the track) correlated with low uptake of the SLP (Fig. 4g, h). Both correlations were disrupted after cell-treatment with CytD (Supplementary Fig. 12B). These observations imply a mechanism where binding events of long duration and high on-membrane mobility are less likely to result in receptor-mediated endocytosis but more likely to enter a cytosolic cross-presentation pathway. Taken together, the correlations between on-cell kinetics and cellular functionality demonstrate the potential of the Glyco-PAINT-APP to uncover functional glycan-structure-activity relationships.

### MR binding on macrophages

Thus far, we have found strong discrepancies in detected glycan binding parameters in the CHO-MR overexpression system versus native lectin-expressing cells. We next studied whether such discrepancies also extended across different myeloid cells. For this, we studied the changes in carbohydrate binding upon macrophage polarization. Certain pro- and anti-inflammatory stimuli are known to affect absolute levels of the MR on the cell surface upon polarisation of the macrophage from its "naïve" M0-like state to either the inflamed M1-like or wound-healing M2-like phenotype[49]. It is, however, not known whether this change in MR (and related CLR) levels also alters binding preferences of the macrophage. We attempted to investigate this by quantifying the binding parameters of mono- and hexavalent mono- and trimannoside 8, 10, 11 and 13 MR ligands, respectively, to murine bone marrow-derived macrophages (BMDM) treated with either M1 inducing (LPS and IFN-γ) or M2-inducing (IL-4) stimuli (Fig. 5a–c). Here, we observed a glycoform-specific trend with increased binding for the hexavalent clusters in comparison to the monovalent clusters, much alike the CHO-MR binding distribution (Supplementary Fig. 4). Additionally, the detected binding events also resulting in endocytic glycan uptake as determined by flow cytometry, where the M2 macrophages appeared to be more endocytic (Supplementary Fig. 13A, B). Surprisingly, a uniform increase in number of binding events (rel. $k_{on}$) across all glycans upon M2 polarization, which is known to upregulate MR expression levels[50], was not apparent from these data. However, for the total binding as quantified by the total track duration (number of events multiplied with their duration) within a square, there appeared to be a vast increase for the M2 phenotype compared to M0 and M1 (Fig. 5c) for all glycans except 13 (when square selection criteria were adjusted to R_Squared > 0.1 to accommodate long duration binding events, as illustrated in Supplementary Fig. 15A–C). This suggests the emergence of a distinct population of long duration binding events that are potentially the result of CLR upregulation by M2 macrophages, analogous to increase in highly mobile long tracks that were seen to be MR-specific on DCs. To evaluate effects of polarization on the spatial clustering of binding events, a clustering analysis was performed using the $x$, $y$ coordinates of each track. This analysis did not result in glycoform-specific trends for clustering of events on macrophage cell surface (Supplementary Fig. 13C).

Next, binding data for polarizing macrophages were pooled with the previously acquired binding data on CHO, CHO-MR, BMDC WT, BMDC MR$^{-/-}$ and BMDM MR$^{-/-}$ to precisely evaluate the contribution of MR-mediated binding events in a single analysis. In Fig. 5d, representative examples of recordings with 13-binding events are shown, and in Fig. 5e the relative contributions of four main kinetic parameters are visualized in radar plots for the combinations of MR-expressing or MR-lacking cell types and BMDM polarization states. From these plots,

it is immediately apparent that indeed the majority of 10 and 13 binding events detected on CHO-MR and BMDM are MR mediated. This contrasts with BMDC, for which there seems to highest degree of redundancy in non-MR binding partners.

## Discussion

This paper describes the single-molecule quantification of glycan-lectin binding on primary immune cells. Whilst quantification of glycan-lectin binding using glycan arrays[51], SPR[52] or ELISA-based[53] technologies with immobilised ligands has yielded a wealth of information related to binding preferences of individual lectins to glycans and vice versa, it has never done so in the context of the living cell surface, where the glycocalyx, other lectins and the constant movement of the membrane can affect glycan binding. Glyco-PAINT-APP offers a significant technological advance that enables the study of glycan-lectin binding in its native context. It is particularly potent, as it now allows the quantification of binding of non-homogeneous distributions of binding events on the cell surface. The method starts with PAINT recordings of glycan binding to heterogeneous, semi-adherent, cell types and can extract kinetic information in the form of off-rates, relative on-rates, diffusion coefficients, displacement and speed of the receptor-ligand interaction in an unbiased and high-throughput manner whilst respecting the subcellular variations typically associated with live cells of primary origin. We believe that the Glyco-PAINT-APP can be of use to expand the information that can be obtained from all PAINT-like technologies, where increasingly attention is shifting to the use of physiological ligands such as peptides, glycans and proteins as imaging probes beyond the original DNA-PAINT method[54–56]. Furthermore, our approach could aid in fundamental investigations into the dynamics of immune cell lectins of other families, such as the Siglecs. It could also be of use for structure-function or target-engagement studies of other ligands that have a typical low affinity interaction with their receptors such as low-affinity antibodies, peptides or TCR-pMHC interactions[57,58], or the other interactions where (weak) binding affinities do not appear to correlate with function, such as the recently reported anti-CD40 antibody library, where affinity reduction led to increased receptor engagement[59].

We acknowledge several limitations of the Glyco-PAINT-APP technology for primary cells that should be considered when interpreting results. First, accurate discrimination between true binding events and background noise remains challenging, making appropriate negative biological controls essential. In the case of myeloid cells, such controls are particularly difficult to obtain, as these cells rarely lack lectin-binding partners due to the high promiscuity of glycan−lectin interactions and the presence of multiple, often weak, binding partners—some of which may not yet be identified. In this context, the CHO/CHO-MR cell line pair provides an optimal single-lectin control, as wild-type CHO cells lack any CLR expression (at least from LC-MS/MS proteomics-based quantification)[60]. Second, lectin trafficking from the plasma membrane to intracellular compartments introduces additional complexity. Although we explored the use of inhibitors such as CytD, the specific effects of these compounds on individual lectins remain largely unknown. Consequently, distinguishing whether the termination of a binding event is due to unbinding or internalization remains inherently difficult. Additional factors that may bias the observed kinetics include degradation of glycan probes by glycolytic enzymes and pH-dependent effects on binding stability. One intriguing aspect of the work is that the live-cell kinetic parameters that were obtained with the Glyco-PAINT-APP approach could be correlated to receptor functions, such as uptake and cross-presentation, which had been the subject of much controversy. The wealth of binding and receptor-ligand information obtained by this approach—which had previously thwarted correlation of bulk binding parameters to function —led us to identify key parameters of the interaction that could be predictive of uptake and cross-

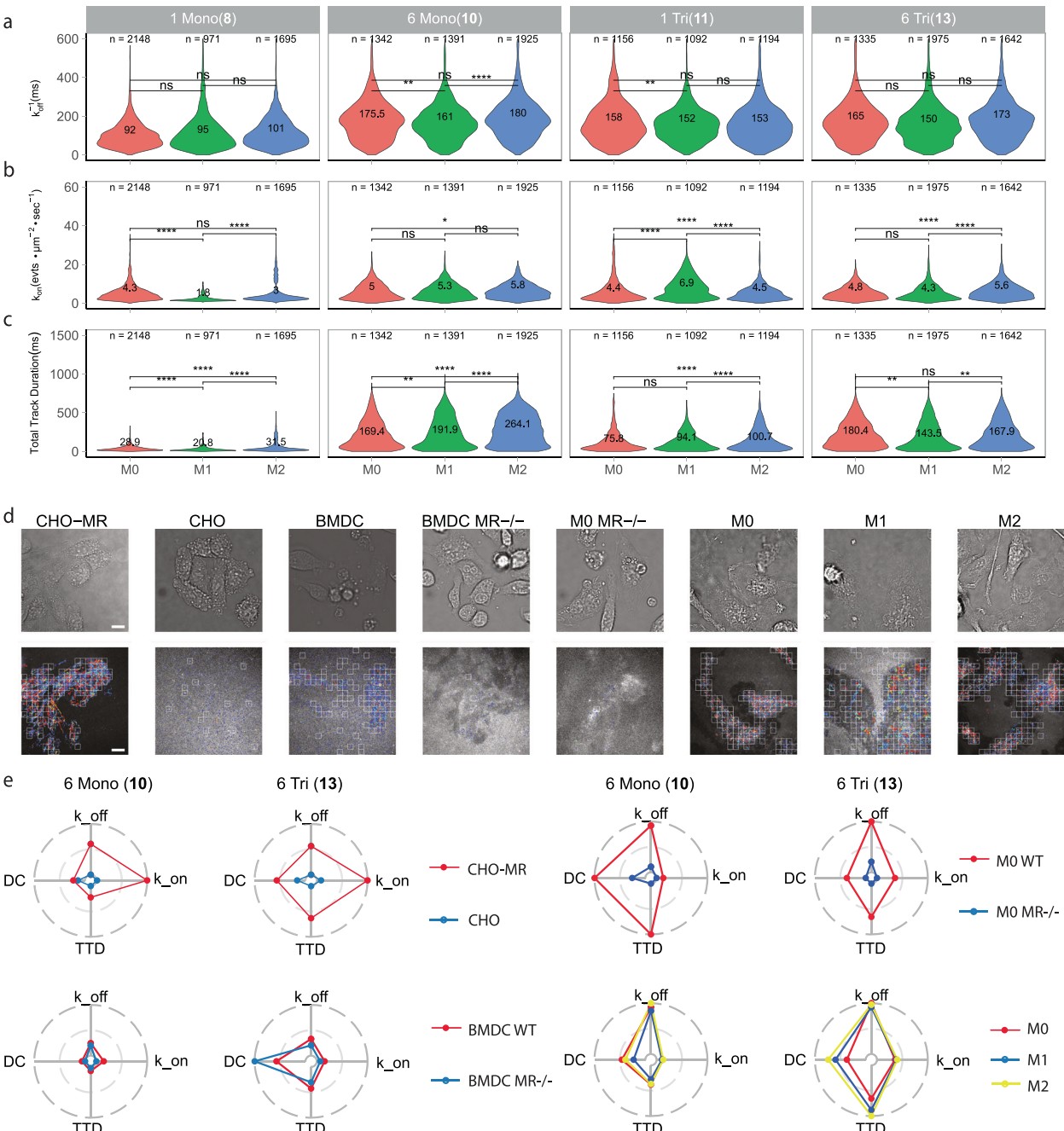

**Fig. 5 | Mannose glycan binding characteristics are highly dependent on lectin expressing cell. a–c** Violin distribution $k_{off}^{-1}$, rel. $k_{on}$ and diffusion coefficient, respectively of glycans 8, 10, 11 and 13 on M0, M1 or M2 polarized macrophages. **d** Representative examples of brightfield images (top row) and track reconstructions (bottom row) of hexavalent trimannoside 13 binding to all cell types that were evaluated in this study. **e** Radar plots summarizing four normalized kinetic parameters for glycan 10 and 13 for the various combinations of cell types. *n*, the number of squares from which the data were obtained. Glyco-PAINT-APP analysis was done using a 20 × 20 grid and filtered with Density_Ratio > 2, R_Squared ≥ 0.9 (for BMDM recordings R_Squared ≥ 0.1) and Nr_Tracks/Square > 20. BMDM record. Significance was assessed using two-way ANOVA followed by a Tukey post-hoc test. Tracks in **c** are coloured from short to long track length (light blue to red). Abbreviations in (**e**) DC Diffusion Coefficient, TTD Total Track Duration, k_on rel $k_{on}$, k_off $k_{off}^{-1}$. Scalebars represent 10 μm.

presentation behaviour: the presence of regions with very long ligand dwell times correlated with cross-presentation ability, and the lack of ligand movement of the receptor with internalization. This—combined with the fact that uptake efficiency and cross presentation showed no correlation—suggests that multiple processes are likely taking place on the cell surface, either mediated by other lectins, or perhaps simply the micropinocytosis of which DCs display a phenomenal rate: they can

internalise their entire cell membrane in about half an hour, or the equivalent of a cell volume in an hour[61,62]. Watts and co-workers have shown macropinocytosis to be a key mechanism for inducing antigen cross-presentation[63,64]. We propose the hypothesis that the increased dwell time on the surface of a DC—either through receptor-mediated interaction or through a-specific interactions—increases the chance of an antigen being internalised in a macropinosome. At the same time,

the binding of a ligand to a clathrin-anchored MR increases its chance of internalisation into a non-cross presentation enabled vesicle. We do, however, need to emphasize that the observed correlations do not by any means implicate a causal relationship. To further explore this causality, experiments with different SLP antigens or genetic knock-outs of proteins other than CRDs involved in the endocytosis pathway could be designed. Our proposed model would, however, explain some of the other reports of non-linear uptake-cross-presentation relationships for mannosylated SLPs[29–31] and potentially even for antibody-antigen conjugates[65].

## Methods

### Mice

Male C57BL/6J and OT-I (C57BL/6-Tg(TcraTcrb)1100Mjb/J) mice were purchased from the Jackson Laboratory (CA, USA), MR[−/−] mice on a C57BL/6 background were originally provided by Lee et al.[66]. The animals were bred and housed under specific pathogen-free conditions and provided with water and food *ad libitum* under a 12:12 day/night cycle. Mice ranging from 8 to 15 weeks old were euthanized by cervical dislocation before harvest of lymphoid organs and/or thigh bones, femur, and tibia.

### Cell culture

**Bone Marrow-Derived Dendritic Cells (BMDC).** Bone marrow (BM) was isolated from femurs, tibias, and thigh bones of WT or MR[−/−] C57Bl/6 J mice via centrifugation (1900 $rcf$, 4.5 min) of scissor-cut bones that were placed in a 1.5 mL tube. The resulting pellet was subjected to red blood cell (RBC) lysis by resuspending in 0.5 mL of ammonium chloride-potassium (ACK) lysis buffer (Gibco, A1049201). After 3 min incubation at rt, the suspension was filtered over a 70 μm filter (Falcon, 352350), rinsed with 5 mL PBS and washed once with PBS by 5 min centrifugation at 300 $rcf$ at rt. Thus obtained BM was cryopreserved in 10% DMSO in FCS according to standard methods[67] or directly resuspended at 1 x 10⁶ cells/mL in 15 cm uncoated culture dishes (Sarstedt, 82.1184.500) in complete RPMI-1640 (Capricorn, RPMI-A) supplemented with 10% heat-inactivated fetal calf serum (FCS, Gibco, A5670701), penicillin (100 I.U./mL) and streptomycin (50 μg/mL) (Gibco, 15140148), 2 mM GlutaMAX (Gibco, 35050061), 50 μM 2-mercaptoethanol (Gibco, 31350010) and 20 ng/mL mGM-CSF (Peprotech, 315-03) and cultured in a humidified incubator at 37 °C and 5% CO₂. On day two, 5 mL of the above-described fresh medium was added, and on day 4, cells were reseeded in fresh medium at 1E6/mL. Cells were used for microscopy and T cell activation experiments on day 7 or 8.

**Bone Marrow-Derived Macrophages (BMDM).** BM was isolated from femurs, tibias, and thigh bones of WT or MR[−/−] C57Bl/6J mice via centrifugation (1900 $rcf$, 4.5 min) of scissor-cut bones that were placed in a 1.5 mL tube. The resulting pellet was subjected to RBC lysis by resuspending in 0.5 mL of ACK lysing buffer. After 3 min incubation at rt, the suspension was filtered over a 70 μm filter, rinsed with 5 mL PBS and washed once with PBS by 5 min centrifugation at 300 $rcf$ at rt. Thus obtained BM was cryopreserved in 10% DMSO in FCS according to standard methods[67] or directly resuspended at 0.8 x 10⁶ cells/mL in 15 cm uncoated culture dishes (Sarstedt, 82.1184.500) in complete RPMI-1640 (Capricorn, RPMI-A) supplemented with 10% heat-inactivated fetal calf serum (FCS, Gibco, A5670701), penicillin (100 I.U./mL) and streptomycin (50 μg/mL) (Gibco, 15140148), 2 mM GlutaMAX (Gibco, 35050061), 50 μM 2-mercaptoethanol (Gibco, 31350010) and 20 ng/mL M-CSF (Biolegend, 576404) and cultured in a humidified incubator at 37 °C and 5% CO₂. On day 2, 5 mL fresh medium was added, and on day 4 medium was aspirated and replenished with 15 mL fresh medium. Cells were used for experiments on day 7 or 8.

**CHO-MR.** The CHO-MR cell line was kindly provided by Luisa Martinez-Pomares[68] and cultured in DMEM/F12 (Capricorn, DMEM-12-A), supplemented with 10% FCS, penicillin (100 I.U./mL), streptomycin (50 μg/mL) and selection antibiotic G418 (0.6 mg/mL). A layer of adherent cells was washed with PBS, and cells were harvested by 10 min incubation with 2 mM EDTA in PBS and subcultured approximately twice per week at a 1:5 split when cells reached 70-80% confluency.

**Statistical analysis and sample size.** Statistical analyses were conducted to compare glycan ligand binding kinetics across probes using the Glyco-PAINT square-based subsampling technology. Subsampling subcellular regions (squares) within fields of view increased the number of independent data points, enhancing statistical power compared to treating entire fields of view or cells as single units. For all Glyco-PAINT experiments, at least 3 biological replicates (independent experiments with fresh mouse material or new passage number for cell lines) with three technical replicates (fields of view per condition) were recorded. For flow cytometry assays, at least three biological replicates with two technical replicates per condition were conducted. A two-way ANOVA was used to assess differences among probes with respect to kinetic parameters derived from Glyco-PAINT experiments and flow cytometric assays, followed by Tukey's Honest Significant Difference (HSD) test for post-hoc comparisons to control Type I error. Significance is reported as: ns (not significant), $p \geq 0.05$; $p < 0.05$ (*); $p < 0.01$ (**); $p < 0.001$ (***); and $p < 0.0001$ (****).

**Fluorescent glycan probes.** Glycan and glycan SLP probes were stored as lyophilized powders at −20 °C. Upon thawing, vials were reconstituted in DMSO and concentration was determined by measurement of absorbance using Nanodrop apparatus with extinction coefficients $e_{sCy5} = 250.000\ M^{-1}cm^{-1}$ at 641 nm and $e_{ATTO655} = 125.000\ M^{-1}cm^{-1}$ at 663 nm. Small aliquots were stored at −20 °C until use.

**OT-I T cell isolation.** A 70 μm filter was placed on a 50 mL tube and pre-wetted with PBS supplemented with 2 mM EDTA and 2% FCS (single-cell suspension buffer, SCSB). Freshly harvested spleens from OT-I transgenic mice were placed on the filter and disrupted with the back end of a syringe. After thorough washing, the suspension was centrifuged (10 min., 300 $rcf$, rt). Next, the pellet was gently resuspended in 2 mL of ACK lysing buffer (Gibco, A1049201). After 3 min incubation, the suspension was diluted with 10 mL PBS, filtered once more, and centrifuged again (300 $rcf$, 10 min, rt). The pellet was resuspended and subjected to magnetically-activated depletion of non-target cells using the "Naïve CD8a+ T cell isolation Kit" (Miltenyi, 130-096-543) according to manufacturer's protocol. The obtained T cells were counted and resuspended at a density of 5-15 x 10⁶/mL in 1 mL PBS with 5 μM CFSE (Biolegend) and incubated for 15 min. at 37 °C. After incubation, cells were spun down (10 min, 300 rcf, rt), washed once more with complete RPMI-1640 and were ready for downstream use.

**Splenic DC isolation.** Spleens isolated from C57Bl/6J mice were placed on a 10 cm petri dish (Sarstedt, 83.3902) containing 5 mL HBSS (Gibco, 14025092) supplemented with 1 mg/mL collagenase IV (Sigma, NC2115693) and 20 U/mL DNAse (Thermo Scientific, EN0525) and minced into small pieces. After incubating for 30 min at 37 °C, tissue digestion was stopped by addition of 2 mM EDTA. The remaining homogenate was disrupted and RBC-lysed as described above and subjected to magnetically-activated depletion of non-target cells using the "Pan Dendritic Cell isolation Kit" (Miltenyi, 130-100-875) according to manufacturer's protocol.

**Glycan uptake and cell surface staining.** 2.5 x 10⁵ WT, MR[−/−] BMDC or BMDM were seeded after treatment with polarizing stimulus as described below in a 96 well v-bottom plate (Sarstedt, 82.1583001).

The next day, Glycan SLP probes were added to the cells at 250 nM and incubated for 1 h at 37 °C. Active uptake was halted by addition of ice-cold PBS and washed twice with PBS + 2% FCS + 2 mM EDTA (300 *rcf*, 5 min., rt). Cells were stained with a selection of the following stains and antibodies: Zombie Yellow (Biolegend, 423103, 1:500), TruStain FcX (Biolegend, cat no 101319, 1:100), CD11c - eFluor450 (clone: N418, eBioscience, cat no 48-0114-82, dilution 1:200), CD206-AF488 or AF647 (clone: MR5D3, Biorad, MCA2235A488T, 1:20), CD86-PerCP (Biolegend, cat no 105025, clone GL-1, 1:200), F4/80-APC-Cy7 (Biolegend, cat no 123117, clone BM8, dilution 1:400), MHC-II-AF488 (Biolegend, cat no 107615, clone M5/114.15.2, dilution 1:1000) for 30 min. on ice. Then cells were washed twice with PBS + 2% FCS + 2 mM EDTA and acquired on Guava EasyCyte 12HT. For BMDM uptake and characterization, an identical procedure was followed but acquisition was performed on Sony ID7000 spectral flow cytometer. Analysis was performed using FlowJo v8. Phagocytic index (PI) was calculated according to equation 1. Statistical analysis and plotting were performed using GraphPad Prism V10.

$$PI = \% \, sCy5^{+} \, cells \times MFI \, sCy5^{+} \, cells - \% \, sCy5^{-} \, cells \times MFI \, sCy5^{-} \, cells$$

(1)

**Confocal imaging.** BMDC were cultured as described above and plated on IbidiTreat μ-Slide 8 Well High dishes (Ibidi, 80806) at 0.5 x 10⁶ cells per well. Cells were incubated with Cy5-labeled SLP 250 nM for 1 h at 37 °C in complete RPMI. Following ligand incubation, cells were washed twice with ice-cold PBS and fixed incubated with CD11c-BV421 (BioLegend, cat no 117329, clone N418, dilution 1:100) antibody for 30 min. on ice. Then cells were washed twice and incubated with 4% paraformaldehyde (PFA) in PBS for 15 min at room temperature. Fixed cells were permeabilized with 0.1% Triton X-100 in PBS for 20 min and blocked with and washed twice with blocking buffer (PBS + 2% FCS + 2 mM EDTA). Then, cells were incubated with LAMP1-AF488 (BioLegend, cat no 121607, clone 1D4B, dilution 1:100) and EEA1-AF594 (MBL Life Sciences, cat no M176-A59, clone 3C10, dilution 1:100) antibodies for 1 h on ice. After final washes, coverslips were mounted in glycerol/DABCO mounting medium. Images were acquired using a Leica Stellaris 8 White Light Laser (WLL) confocal microscope equipped with a 63×/1.40 NA oil-immersion objective. Excitation was performed at 638 nm for Cy5, 488 nm for AF488 and 561 nm for AF594, with detection windows set according to manufacturer recommendations to avoid channel crosstalk. Line averaging ($n = 2–4$) and sequential scanning were applied where appropriate. All acquisition parameters (laser power, detector gain, pinhole size) were kept constant between conditions. Images were processed in LAS X software (Leica Microsystems) and analyzed in Fiji (ImageJ) without nonlinear contrast adjustments.

**T cell proliferation assay.** 5 x 10⁴ WT, MR⁻ᐟ⁻ BMDC or 1 x 10⁵ splenic DC were seeded in 96-well Nunc U-bottom plates (Thermo Scientific, 168136) and pulsed with Glycan SLP at 40 nM for indicated time. In indicated experiments, DC were pretreated with 1 μg/mL MPLA (Avanti, 699800 P) for 2 h before antigen pulse. After antigen pulse was finished, the plate was spun down for 3 min at 600 *rcf*, washed once with complete RPMI-1640 and 1-1.5E5 freshly isolated, CFSE-stained OT-I T cells were added to the pulsed DCs in 200 μL complete RPMI-1640 and incubated at 37 °C. After 3 d of coculture, the plate was spun down (3 min, 600 rcf, 4 °C) and supernatant was removed and stored at −80 °C. Cells were washed with FACS buffer (PBS with 2 mM EDTA, 2% FCS and 7.4 mM NaN₃) and stained with Aqua Live/DEAD (Invitrogen, L34957, 1:500), TruStain FcX (Biolegend, 101319, 1:100), CD8a−APC (clone: 53-7.6, Biolegend, 100711, 1:200), and TCR V beta 5.1/5.2−eFluor450 (clone: MR9-4, eBioscience, 48-5796-82) for 30 min on ice, washed two times and acquired on a BD Fortessa I flow cytometer and analyzed using FlowJo v8. To calculate mean cycle, the CFSE dilution factor was obtained by dividing the MFI of the antigen

pulsed condition by the DMSO pulsed condition expressed as Log2[69]. Statistical analysis and plotting were performed using GraphPad Prism V10.

**BMDM polarization.** At day 6, adherent macrophages were harvested by aspiration of medium, washing once with PBS and incubation with 10 mL of 2 mM EDTA in PBS for 10 min at 37 °C. Cells were reseeded in complete medium with addition of polarizing stimulus, 20 ng/mL IFN-γ (Peprotech, 315-05-100UG) + 100 ng/mL LPS-EB (Invivogen, tlrl-eblps) for M1 or 20 ng/mL IL-4 (Peprotech, 214-14-20UG) for M2 for 16 h.

**Glyco-PAINT optical setup.** Single molecule imaging was performed on a Nikon Ti2 N-STORM system equipped with TIRF module, Z piezo element, perfect focus system for axial drift correction and an OkoLab incubator with temperature and CO2 controller (37 °C and 5% $CO_2$) for live-cell imaging. Recordings were acquired using the 647 nm excitation laser (160 mW, 1.9 kW/cm²). Upon laser excitation, fluorescence was collected by a 100x 1.49 NA oil-immersion objective, passed through a quad-band dichroic mirror (97335 Nikon), and detected by a Hamamatsu ORCA Flash 4.0 CMOS camera with 160 nm pixel size. The signal was collected using the following settings: 512 × 512 pixel region, no binning, pixel depth 16-bit, exposure time 50 ms, live-cell observation and 2D-STORM (lens out), zoom 1x, lens x0.4, and for live-cell observation at 37 °C, the correction collar was set to position 8160.

**Acquisition of Glyco-PAINT recordings.** 5 x 10⁴ CHO-MR or 1 x 10⁵ WT, MR⁻ᐟ⁻ BMDC or BMDM were seeded in 8-well glass-bottomed microscopy slides (Ibidi, 80827) in complete medium. After equilibration in the microscope incubator, fluorescent glycan was added at 5 nM for CHO-MR and 10 nM for BMDC or BMDM experiments. Then, cells were brought into focus using brightfield illumination and 2000 frames (at 50 ms intervals) were recorded within a single field-of-view at 40−60% of maximum 647 nm laser power using TIRF illumination. For indicated experiments, Cytochalasin D (Focus Biomolecules) was added to a final concentration of 10 μM for 30 min prior to acquisition.

**ROI-based analysis of Glyco-PAINT recordings.** TrackMate was run manually using the Fiji plugin[70]. The LoG spot detection algorithm was applied with an object radius of 0.5 μm, with pre-processing with median filter and sub-pixel localization unchecked. Threshold values were set to 5, 10 or 15 such that no more than 1.500.000 spots were detected and kept identical per experimental condition. Next, single-particle tracking was performed using the Simple LAP tracker algorithm with a maximum frame gap of 3, a max linking distance of 0.6 μm and a gap closing max distance of 1.2 μm. Tracks with only two spots were discarded. Then, manual Regions of Interest (ROIs) were drawn in individual recordings using Fiji based on brightfield-defined cell outlines. The number of tracks residing within the ROI was compared with the number of tracks outside the ROI to establish a ratio of cell/glass density. Only recordings for which that ratio exceeded 2 were considered for further analysis. For all remaining recordings, the $k_{on}$, $k_{off}$ and MSD were calculated for tracks within the ROI according to the procedure as described by Riera et al.[10]. Statistical analysis and plotting were performed using GraphPad Prism v10.

**Glyco-PAINT-APP analysis of recordings.** Processing of recordings using Glyco-PAINT-APP was performed as described in a step-by-step procedure in the Supplementary Software Manual in the Supplementary Information and accompanying Supplementary Videos 1 and 2. For the TrackMate processing, a batch file (Experiment Info.csv) containing the experiment metadata and tracking parameters was created. Threshold values were set to 5, 10 or 15 such that no more than 1.500.000 spots were detected. Recordings were then processed in TrackMate using the 'Run TrackMate Batch' plugin provided by the Glyco-PAINT-APP. Through this plugin, spot detection and tracking by

TrackMate[70] was performed as indicated in the batch file using the Simple LAP tracker algorithm with a maximum frame gap of 3, a max linking distance of 0.6 μm and a gap closing max distance of 1.2 μm. Tracks with only two spots were discarded. For the parameter sensitivity analysis, each tracking or spot detection was varied as indicated, whilst the others were kept at aforementioned values (Basis scenario).

With the Glyco-PAINT-APP utility 'Generate Squares', a grid of squares was overlaid, and kinetic properties for each square were calculated. Default parameters for grid processing are (deviations are mentioned in figure captions): Nr of Squares in row 20, Minimum Tracks to Calculate Tau 20, Min allowable R Squared 0.1, Min Required Density Ratio 2, Maximum Allowable Variability 10 and Neighbour Mode Free.

For every recording, a background track count was calculated by averaging the track count of the 40 (10% of the total number of squares) least dense squares. Only squares for which the track count exceeded the Min Required Density Ratio of 2 were considered (and the remaining ignored as background). For each square, the variability was calculated and only squares for which the variability was less than the Maximum Allowable Variability of 10 were considered. For squares meeting both the Minimum Required Density Ratio and Maximum Allowable Variability criteria, and containing at least the Minimum Tracks to Calculate Tau kinetic parameters, including $k_{on}$, $k_{off}$ and MSD were calculated as in Riera et al.[10], or copied from the TrackMate Tracks table output (for velocity, displacement, and track duration).

Summary files were created using the "Compile Project" utility, creating an "All Recordings.csv" file, containing information for all 1119 recordings, an "All Squares.csv" file containing information on 389,200 ($973 \times 400$) squares and an "All Tracks.csv" file containing information on approximately 16 million tracks (binding events) in the project. Statistical analysis and plotting using these merged files were performed using the ggplot2 package in R[41,71].

Spatial clustering of single-molecule tracks was quantified as $L(r) - r$ at $r = 2$ μm from Ripley's $L$-function, computed using spatstat.explore and spatstat.geom packages in R on filtered $x/y$ track coordinates from pre-selected squares, with differences between adjuvant conditions assessed by pairwise Wilcoxon tests with Benjamini–Hochberg correction.

### Ethical statement
All animal experiments received approval from the Dutch Central Authority for Scientific Procedures on Animals (CCD) on license number AVD1060020198832 and were conducted in accordance with the European Union Directive 2010/63/EU, recommendation 2007/526/EC.

### Reporting summary
Further information on research design is available in the Nature Portfolio Reporting Summary linked to this article.

## Data availability
The raw single-molecule microscopy datasets generated during this study (~1 TB total) are available from the corresponding author upon request due to file size limitations. Representative raw recordings and all processed microscopy tracking data have been deposited in the Zenodo repository under DOI: [https://doi.org/10.5281/zenodo.17485662]. All quantitative data supporting the findings, including square-level parameters and flow cytometry MFI values, are provided in the Source Data file and on Zenodo. Source data are provided with this paper.

## Code availability
The Glyco-PAINT-APP software package is available on GitHub: [https://github.com/Leiden-chemical-immunology/Glyco-PAINT] and will be subject to continuous improvement, a frozen version at the time of submission is available via Zenodo DOI above. Installation instructions, demonstration datasets and a software manual are also available via GitHub.

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

## Acknowledgements

We thank the Flow Cytometry Core Facility (FCF) of Leiden University Medical Center (LUMC) in Leiden, The Netherlands for experimental support, use and maintenance of BD Fortessa I and Sony ID7000. We thank the Animal Research Facility (ARF) and Bram Slütter at the Leiden University Faculty of Science for mouse breeding and colony management. We thank Hans van Elst and Nico Meeuwenoord for assistance with HPLC and SEC purifications. This work was funded by ZonMW, and the European Research Council (ERC-CoG 865175-KineTic) awarded to SIvK.

## Author contributions

K.S.: Conceptualization, Methodology, Software, Formal Analysis, Investigation, Resources, Data Curation, Writing—Original Draft, Visualization. J.J.A.B.: Conceptualization, methodology, software, formal analysis, investigation, resources, data curation, writing—review and editing. W.D.: Conceptualization, methodology, investigation, resources, writing—review and editing. D.T.G.: Methodology, investigation, writing—review and editing. A.C.: Methodology. C.K.: Conceptualization, resources. RR: Conceptualization, software. L.A.: Conceptualization, writing—review and editing. S.Iv.K.: Conceptualization, supervision, writing—review and editing, funding acquisition.

## Competing interests

The authors declare no competing interests.
