## [Transparent Peer Review file · Nature Communications]

Subcellular glycan-mannose receptor binding kinetics correlate with myeloid cell function

Corresponding Author: Professor Sander van Kasteren

Version 0:

Reviewer comments:

Reviewer #1

(Remarks to the Author)

The authors propose a technique to assess glycan-lectin interactions at the subcellular level in live cells, which is based on the collection of a series of fluorescence images (using fluorescently tagged carbohydrates) and their post processing. Interestingly, using the selected cell types that the authors study, the results show a non-uniform distribution of binding events at the cell surface and the authors claim that, using this technique, it is possible to quantify the kinetics of the detected interactions, namely, on and off rates, as well as diffusion coefficients of the binding complexes on the surface of individual cells. Overall, the concept of the manuscript is appealing, and the proposed methodology has conditions to be used by the scientific community. However, there are two main concerns related to this manuscript that I would like to point out:

1) A proper validation of the methodology would require the authors to compare their results to known/established methods. I understand that in the context of the work, it might be difficult to find adequate data or techniques that could be used as a reference. However, as the authors mention, there are other techniques that can quantify the glycan-lectin interactions, as for example SPR, ELISA, etc. While these techniques are not the same as doing the experiments in live cells (as the authors propose here), the biophysical parameters (on, off rates, etc.) should not be dramatically different. I recommend that the authors compare their results with the ones obtained from those techniques to validate their methodology.

2) The results are interesting, showing some surprising data related with the mobility of the glycan/lectin complexes and a correlation with internalization or their cross-presentation on the cell surface. The ability to do this type of assessment is important and challenging to do with the current methodologies, supporting the relevance of the manuscript. However, the work, as it is presented, focus more on a methodological perspective and less in the study of a particular biological/biophysical system, leading to my belief that the work would be more suitable for the methodological journal that focus on the publication of protocols.

Minor Comments:

- 1) CTL is mentioned for the first time in line 56, and DC in line 126, but neither has been defined previously. Please correct.
- 2) Line 84, reformulate the phrase.
- 3) Line 143, check the units at the end of the line
- 4) Please check the whole text for grammar and typos

(Remarks on code availability)

Reviewer #2

(Remarks to the Author)

Comments for authors

The paper by Steuten et al., "Quantification of single molecule glycan-mannose receptor binding kinetics on myeloid cells reveals high subcellular binding heterogeneity" introduces Glyco-PAINT-APP. This is an automated processing pipeline aimed at quantifying glycan binding dynamics at the subcellular level using a PAINT-based single-molecule imaging technique. The authors demonstrate its application in profiling glycan-lectin binding analysis in immune cells, particularly

dendritic cells and macrophages, and correlate these interactions to immune receptor function. The new methodology enhances existing Glyco-PAINT techniques by implementing an automated analysis workflow, improved spatial resolution and therefore enabling biological function correlation.

This is one of the first automated high-resolution analyses of glycan-lectin interactions. The study successfully maps glycan binding event density, dwell times, and diffusion coefficients at the subcellular level allows new insights into receptor biology what is quite exciting.

Overall, the paper is well written, and the work is technically sound and well-structured, but several issues must be clarified to strengthen the study conclusions and ensure a more robust interpretation of the data.

Here are the points:

1. For the single-particle tracking, the authors used the 'Simple LAP tracker algorithm with a maximum frame gap of 3, a max linking distance of 0.6 μm , and a gap closing max distance of 1.2 μm . Tracks with only two spots were discarded' (lines 1022, 1038). Can these parameters be considered reliable across all experimental conditions? Did the authors benchmark the LAP tracker against alternative segmentation or tracking methods (e.g., Otsu thresholding, machine learning-based segmentation) to validate its robustness? To what extent do these LAP tracking parameters require optimization based on fluorophore brightness, molecular mobility, or frame rate? Was any experimental or manual validation performed to ensure tracking accuracy under the specific conditions used in this study?

2. Spot detection is critical in the Glyco-PAINT-APP study, as accurate identification of glycan binding events directly impacts tracking reliability and the resulting kinetic measurements. Did the authors manually verify a subset of detected single-molecule events and compare them to the automated output? Additionally, were small changes in detection threshold tested for their effect on event counts or kinetics? The authors mention using thresholds of 5, 10, and 15, but a direct comparison of the resulting datasets is not provided. Since threshold selection can introduce detection bias - either by excluding weak but valid events or by including noise - it is important to explicitly assess the robustness of the analysis across different threshold settings and address potential bias in spot detection.

3. Regarding the figure 2: What kind of "select squares criteria"—free or strict—was chosen in b and c? After watching the videos, the images confuse me a bit. It seems like in figure 2b, the square areas are quite noisy with high background in comparison to 2f. Was the same background correction applied to both images, or were different noise reduction methods used?

Additionally, the squares represented in g and h do not correspond to the squares in f; rather, they are additional squares in different positions. Why were different squares analyzed instead of using the same ones as in f? Wouldn't it be more consistent to show and analyze the same squares (in f, g and h) as was done for the Trimannose Cluster (13) analysis? If different squares were intentionally chosen in f, g and h, what was the rationale for this selection? Given that the square selection method can significantly influence the interpretation of spatial binding behavior, it is important to clarify the selection criteria, ensure consistent background processing, and maintain comparability across panels.

4. Regarding figure 4 and figure S7, where the binding parameters of glycosylated SLPs to cells were tested: The authors stated (lines 277-280) that the on-rates, off-rates, and diffusion coefficients were significantly increased for all glycosylated probes (1-6) but not for the non-glycosylated control (7). However, this does not fully align with the data presented in Figure S7.

Specifically, in the CHO-MR treated condition (figure S7, g and h) and the BMDG-treated condition (figure S7, a and c), on-rates and off-rates do not consistently increase - in some cases, they partially decrease, contradicting the claim of a uniform increase. Additionally, the on-rate for the non-glycosylated control antigen 7 also increases in certain conditions (figure S7, c, d, and h), which is inconsistent with the statement that only glycosylated probes exhibit this effect.

Please comment on the discrepancy between the reported binding kinetics in figures 4 and S7 and the expected trends based on previous studies. Studies on mannose receptor (MR) and DC-SIGN binding kinetics (Feinberg et al., 2001; Taylor et al., 1992) suggest that trimannosylated ligands should exhibit lower off-rates than monomannosylated ones due to multivalent binding effects. However, figure 4 shows no such difference. Additionally, prior work on actin-dependent receptor dynamics (Lau et al., 2007) indicates that Cytochalasin D should disrupt clustering and reduce binding stability, rather than uniformly increasing on- and off-rates. Can the authors clarify how their findings align with these previously reported trends?

5. The manuscript uses different Density_Ratio thresholds (>10 in case of macrophages and >2 for BMDCs). Can the authors clarify why different thresholds were chosen and whether these values were optimized based on a specific criterion? How does varying the Density_Ratio threshold affect the interpretation of the results? A direct comparison of results using different Density_Ratio values would help assess the robustness of the findings and eliminate bias in data interpretation.

6. Regarding figure 5: In figure 5b, the square areas appear quite noisy with a high background compared to Figures 5a and 5c. This raises concerns about the high Density_Ratio threshold used for this dataset.

Would it be possible that the polarization towards M1 macrophages primarily detects strong clustering or receptor hotspots, rather than capturing the full range of receptor behavior? Would broader changes in receptor behavior, including moderate enrichment, be more effectively detected using Density_Ratio >2 instead of >10?

A Density_Ratio >2 might be more suitable for analyzing broader spatial patterns, allowing for more variation in binding

behavior to be captured, rather than focusing exclusively on the most intense binding regions.

Can the authors comment on this? I may be misinterpreting the reasoning behind the choice of Density_Ratio thresholds, so clarification on whether different thresholds influence the biological conclusions would be valuable.

7. Additionally, the figure 5 provides interesting insights into how macrophage polarization affects MR ligand binding, particularly with hexatrimannoside. However, testing only one ligand does not fully establish whether polarization alters MR binding specificity. Would testing additional glycan ligands (e.g., mono- and trimannosylated ligands) help determine whether MR binding preferences shift in M2 macrophages?

Additionally, could receptor clustering contribute to prolonged dwell times in M2 macrophages rather than just increased MR expression? A fluorescence-based receptor clustering analysis could clarify whether M2 macrophages exhibit larger receptor aggregates, leading to longer binding events.

The authors also state that binding events in the M2 state exhibited longer diffusion coefficients compared to the M1 or M0 state. However, from the data, the log diffusion coefficient appears nearly the same between M1 and M2 macrophages, with only the frequency of events increasing. This suggests that binding kinetics are altered by the number of MR-positive events rather than by an inherent difference in receptor diffusion behavior. Can the authors clarify this point?

Furthermore, in Figure 5B, it is noticeable that some tracks exhibit long track lengths, but these tracks are unfortunately not covered by the grids used in the analysis. Could this influence the interpretation of the results? If long tracks are systematically excluded from the analysis, the impact of extended receptor-ligand interactions may be underestimated. Would an alternative grid size or adaptive segmentation approach help capture these events more effectively? Lastly, since M1 macrophages exhibit increased receptor shuttling, would comparing MR internalization rates clarify whether ligand binding is affected by receptor turnover rather than diffusion alone?

8. For a balanced interpretation of the data, the discussion section should more transparently address study limitations. In particular, potential confounding variables such as receptor trafficking, glycan degradation, and pH effects need to be considered, as they could influence the observed binding kinetics. Additionally, the authors should explicitly acknowledge that correlation does not automatically imply causation, and alternative explanations for the reported effects should be discussed.

(Remarks on code availability)

The GlycoPaint-Pipeline repository provides a structured and well-documented analysis pipeline for single-molecule glycan-receptor interaction data. The authors supply detailed installation instructions through a dedicated Installation.md file, guiding users through the setup of Fiji, TrackMate, Python, and R environments. The codebase includes both Python and R scripts and is organized with clear folder structures and metadata.

Importantly, the repository also includes test datasets (Demo/ folder), such as All Tracks.csv and Experiment Info.csv, which allow users to verify installation and explore example outputs. This is a valuable addition that enhances the reproducibility of the pipeline.

That said, I found the large number of files and subfolders a bit overwhelming to navigate, and it's possible I missed some details despite reviewing the materials thoroughly. While I'm not a programming expert, I do have more experience than a typical user, and from that perspective, the pipeline is moderately accessible. However, the overall usability would improve with better integration between the data, code, and documentation to make it easier to understand how the components relate and how to get started.

To support broader use and reproducibility, the authors are encouraged to:

- Add a walkthrough or script that demonstrates how to use the demo dataset to generate key outputs or figures,
- Clearly link specific scripts to corresponding analyses or figures in the manuscript,
- Providing a requirements.txt or environment file to streamline setup.

Overall, the pipeline is a valuable and technically well-executed resource for specialized users and offers a strong foundation for reproducible glycan-binding analysis.

Reviewer #3

(Remarks to the Author)

(Remarks on code availability)

Reviewer #4

(Remarks to the Author)

The authors describe a very interesting development of the Glyco-paint method to discriminate in a space resolved manner different kinetics of binding of glycans to myeloid cells. The pipeline analysis allowing glycans analysis is of high potential impact for the study of cells-glycan interactions. Nevertheless, there is a lack of evidence to suggest that the binding observed is on Mannose Receptor and not a mixture of different C-Type lectin receptors on the cells. Similarly, the uptake of glycoclusters is not sufficiently supported.

Main comments

The authors mention the specific Mannose receptor in their study but other CLRs are able to recognize mannoses (like DC-SIGN for example), especially clustered mannose that might induce a lot of binding to other lectins. Is it possible to confirm that the binding observed is only on MR through labelling of endogenous MR (with antibody?) or MR knockout? Otherwise, the manuscript should be altered to reflect that the lectin binders are unknown.

Can the heterogeneity of binding observed be ascribed to the binding to different lectins?

The experiments on BMDC and CHO were performed with different concentrations of glycan probes, respectively 5 and 1nM. Could you comment this difference?

The uptake of the SLP-glycomolecules is measured by flow cytometry, which could also show binding to the cell surface. It would be interesting to show by confocal microscopy that the SLP-molecules are located in vesicles inside the cells to confirm the uptake.

Minor comments

Page 5 lines 140-147. Please explicit why the early observations on BMDC are surprising in more detail as it is the basis of the study and will help the reader to understand the questions asked.

Could you also explicit which probes were used here and the rationale behind this probe choice. Figure 3 showing the structures of the fluorescent sugars is introduced only later.

Figure 1 c) data inspection and curation figure is too small to be informative. The same applies to the schemes of resting and polarised cells that is maybe a bit oversimplistic. It would be helpful for the reader to introduce the structure of the glycan probes and parts of the averaged data to show the rationale behind the study (fig S1 E-F).

It is surprising on Fig S1 F that the difference in koff between glycans 1 and 6 is not judged significant.

Fig 3 b,c please move labels (eg 1tri) so they are visible.

(Remarks on code availability)

Reviewer #5

(Remarks to the Author)

- What are the noteworthy results?

In the manuscript titled "Quantification of single molecule glycan-mannose receptor binding kinetics on myeloid cells reveals high subcellular binding heterogeneity" by Kas Steuten, Johannes J.A. Bakker, Ward Doelman, Diana Torres-García, Roger Riera, Lorenzo Albertazzi and Sander I. van Kasteren, the authors

describe an automated image processing pipeline they refer to as Glyco-Paint-APP that aims to extract sub-cellular glycan interaction kinetics. The work describes a spatially resolved quantitative image analysis approach that is applied to a method described in a previous study by Riera et al. (2021) Nature Chem. Biol. titled "Single-Molecule Imaging of Glycan-Lectin Interactions on Cells with Glyco-PAINT". The application of their pipeline allows them to quantify carbohydrate lectin binding kinetics based with single molecule resolution.

While I am not an expert in the field of myeloid cells or lectin binding, I find the method description for quantitative spatially resolved kinetic measure of glycan interaction with receptors from glyco-PAINT experiment to be interesting and of wider relevance. Steuten et al. demonstrate using the image-analysis pipeline developed in FIJI with interface to TrackMate (Glyco-PAINT-APP), that receptor kinetics are indeed spatially inhomogeneous. Thus the technical development and the finding both form novel aspects of their work, improving on previous approaches. Using glycosylated synthetic long peptides (SLPs) they quantify immune cell uptake of SLPs and the role of actin polymerization. They find the residence time of glycan-SLPs to be the best positive predictor of cross-presentation in dendritic cell co-culture with CD8 T-cells. They also find the on-rates and diffusion coefficients in carbohydrate binding of mannose receptors (MR) correlate with polarization state of macrophages. Taken together these results suggest a utility for sub-cellular analysis using the tools developed for immune-cell dynamics and the role of carbohydrate ligand-binding kinetics.

- Will the work be of significance to the field and related fields? How does it compare to the established literature? If the work is not original, please provide relevant references.

The application of a pipeline for single molecule tracking and extracting binding and mobility kinetics from STORM microscopy is likely indeed to have wider impact beyond the cellular systems described. The use of such a tool through standard FIJI plugins is likely to allow for its wider use.

- Does the work support the conclusions and claims, or is additional evidence needed?

The claims made through the manuscript indeed are supported by sufficient data. Some minor changes in the manuscript that would help clarity of representation of the data are added.

- Are there any flaws in the data analysis, interpretation and conclusions? - Do these prohibit publication or require revision? This reviewer has noted minor changes that could help make some of the figures clearer. These need to be addressed in order to support the manuscript's claims.

1) Figure 1c: Please add units to the diffusion coefficient and track duration plots. Unless they are normalized, in which case

the legend needs to clarify this.

2) Figure 1 plots appear to be screenshots of the outputs and the software interface and could be improved upon with high resolution figures.

3) Multiple figure legends have font sizes that are quite variable making them at times hard to read (e.g. in Fig. 4).

4) The brightfield images in Fig. 5 lack contrast. It might help to outline the features that the authors wish to highlight by outlining cells, similar to what has been done by them in Fig. 1.

- Is the methodology sound? Does the work meet the expected standards in your field?

The methodology adopted is based on fairly standard algorithms of single particle tracking implemented in TrackMate under FIJI and used by the authors. The approach taken is a good combination of existing ideas applied to a novel problem.

- Is there enough detail provided in the methods for the work to be reproduced?

Yes

(Remarks on code availability)

I found the instructions run and install the script somewhat confusing beginning with system configuration, paths and getting the FIJI plugin to work. The authors could clarify the installation procedure for usability of their code.

I have not reviewed the code line-by-line since I was unable to actually run it.

Version 1:

Reviewer comments:

Reviewer #1

(Remarks to the Author)

The authors followed most of the recommendations from the reviewers. Importantly, the included modifications significantly increased the clarity of the discussion and conclusions. The recommendations that the authors decided not to follow are well justified in the comments to the reviewers. Overall, the quality of the revised version of the manuscript, when compared to the original version, has significantly improved and, in my opinion, the present version is at a scientific level compatible with the high standards of Nature Communications.

(Remarks on code availability)

Reviewer #2

(Remarks to the Author)

Dear Authors,

Thank you for your thoughtful and well-structured rebuttal letter, and for the substantial revisions made to your manuscript. I appreciate the care with which you addressed each of the reviewer points.

Your parameter sensitivity analyses, additional experiments on macrophage polarization, the clarification of figure inconsistencies, and the improvements in both the manuscript text and supplementary material have significantly strengthened the work. I particularly welcome your transparent acknowledgment of limitations (e.g., interpretation of diffusion kinetics, MR clustering assumptions, and correlation vs causation). The additional data (e.g., MR-knockout controls, density ratio comparisons, and extended glycan panel) enhance the robustness and biological relevance of your findings.

While the processing pipeline remains technically complex, the revised documentation now offers much clearer guidance and structure, and the study represents a valuable contribution to the field of single-molecule glycan analysis.

Thank you for your responsiveness and efforts to improve the quality of the manuscript. I have no remaining major concerns.

(Remarks on code availability)

I have re-reviewed the Glyco-PAINT Application Processing Pipeline based on the updated supplementary material. The authors have made a commendable effort to clarify and improve the usability, reproducibility, and transparency of their code. From my perspective as a more advanced user (but not a developer), the pipeline is now clearly documented and technically sound, although it remains complex. The documentation is significantly improved, and I believe the authors have fully addressed the previous concerns.

Reviewer #3

(Remarks to the Author)

(Remarks on code availability)

I co-reviewed this manuscript with one of the reviewers who provided the listed reports. This is part of the Nature Communications initiative to facilitate training in peer review and to provide appropriate recognition for Early Career

Researchers who co-review manuscripts.

Reviewer #4

(Remarks to the Author)

The authors have correctly answered to the questions raised during the first round of review.

(Remarks on code availability)

Please find attached our revised manuscript titled '*Quantification of single molecule glycan-mannose receptor binding kinetics on myeloid cells reveals high subcellular binding heterogeneity*'. We want to thank the reviewers for the detailed and thorough review of the paper and have as a result significantly altered the manuscript, including multiple additional experiments that we have detailed in a point-by-point below.

Yours sincerely,

Sander I. van Kasteren, on behalf of all co-authors

Point-by-Point Response to Reviewer's Comments

Reviewer #1 (Remarks to the Author):

The authors propose a technique to assess glycan-lectin interactions at the subcellular level in live cells, which is based on the collection of a series of fluorescence images (using fluorescently tagged carbohydrates) and their post processing. Interestingly, using the selected cell types that the authors study, the results show a non-uniform distribution of binding events at the cell surface and the authors claim that, using this technique, it is possible to quantify the kinetics of the detected interactions, namely, on and off rates, as well as diffusion coefficients of the binding complexes on the surface of individual cells. Overall, the concept of the manuscript is appealing, and the proposed methodology has conditions to be used by the scientific community. However, there are two main concerns related to this manuscript that I would like to point out:

We thank the reviewer for the positive assessment of our work. We will address each of the comments in the following paragraphs.

1) A proper validation of the methodology would require the authors to compare their results to known/established methods. I understand that in the context of the work, it might be difficult to find adequate data or techniques that could be used as a reference. However, as the authors mention, there are other techniques that can quantify the glycan-lectin interactions, as for example SPR, ELISA, etc. While these techniques are not the same as doing the experiments in live cells (as the authors propose here), the biophysical parameters (on, off rates, etc.) should not be dramatically different. I recommend that the authors compare their results with the ones obtained from those techniques to validate their methodology.

We apologise for not mentioning the fact that we had previously done this comparison in the first Glyco-PAINT manuscript (<https://pubmed.ncbi.nlm.nih.gov/34764473/>). Here we found that the

binding by SPR and Glyco-PAINT on the CHO cell lines correlated relatively well. However, as the current study extends this approach to primary cells with native receptor expression, we have taken the binding data of the CHO-MR cells as the benchmark, rather than re-benchmarking with SPR for the comparison of the primary cell lectin binding, as the complexity of the glycan-binding environment—including heterogeneous lectin repertoires and potential cis-ligand interactions – already gave such big differences, that we focused on solving those.

2) The results are interesting, showing some surprising data related with the mobility of the glycan/lectin complexes and a correlation with internalization or their cross-presentation on the cell surface. The ability to do this type of assessment is important and challenging to do with the current methodologies, supporting the relevance of the manuscript. However, the work, as it is presented, focus more on a methodological perspective and less in the study of a particular biological/biophysical system, leading to my belief that the work would be more suitable for the methodological journal that focus on the publication of protocols.

We agree with the assessment of the reviewer that the first iteration of the paper was 'heavy on the methodology'. Whereas we think that the method an sich is of high value to the scientific community, we have placed extra emphasis on the biological findings this new method has allowed us to obtain. For this, we have performed extra experiments with MR-knockout DCs, to more clearly tease apart the correlation between binding parameters, uptake and cross-presentation, in a manner not accessible through existing approaches. Figure 4A, B and 5E show that specific binding sub-populations on the DCs and macrophages are MR-specific events, thereby for the first time identifying specific lectin binding within a complex 'lectinome' (apologies for the neologism) on a primary cell and correlating these specific binding events to the functions ascribed to this receptor.

Minor Comments:

- 1) CTL is mentioned for the first time in line 56, and DC in line 126, but neither has been defined previously. Please correct.
- 2) Line 84, reformulate the phrase.
- 3) Line 143, check the units at the end of the line
- 4) Please check the whole text for grammar and typos

All corrections agreed and adapted in revised manuscript and can be seen in the marked manuscript.

Reviewer #2 (Remarks to the Author):

The paper by Steuten et al., "Quantification of single molecule glycan-mannose receptor binding kinetics on myeloid cells reveals high subcellular binding heterogeneity" introduces Glyco-PAINT-APP. This is an automated processing pipeline aimed at quantifying glycan binding dynamics at the subcellular level using a PAINT-based single-molecule imaging technique. The authors demonstrate its application in profiling glycan-lectin binding analysis in immune cells, particularly dendritic cells and macrophages, and correlate these interactions

to immune receptor function. The new methodology enhances existing Glyco-PAINT techniques by implementing an automated analysis workflow, improved spatial resolution and therefore enabling biological function correlation.

This is one of the first automated high-resolution analyses of glycan-lectin interactions. The study successfully maps glycan binding event density, dwell times, and diffusion coefficients at the subcellular level allows new insights into receptor biology what is quite exciting. Overall, the paper is well written, and the work is technically sound and well-structured, but several issues must be clarified to strengthen the study conclusions and ensure a more robust interpretation of the data.

We thank the reviewer for the positive assessment of our work. We will address each of the comments in the following paragraphs.

For the single-particle tracking, the authors used the 'Simple LAP tracker algorithm with a maximum frame gap of 3, a max linking distance of 0.6 μm , and a gap closing max distance of 1.2 μm . Tracks with only two spots were discarded' (lines 1022, 1038). Can these parameters be considered reliable across all experimental conditions?

*Yes, they can. We have performed a sweep of LAP tracker parameters to determine variance of the results with different tracking parameters and found that the chosen values corresponding to the 'basis scenario' (and as cited by the reviewer) were able to effectively discriminate MR-mediated binding events from background in the CHO-MR vs CHO cell line pair (see figure S5A, B). When deviating from the 'basis scenario' parameter set, it appeared that mostly the minimum number of spots for a track had effects on the separation between CHO-MR and CHO-WT squares. The new data underpinning these findings have been included as Figure S5-6 and as additional text under subheading **Glyco-PAINT-APP** of the main text.*

Did the authors benchmark the LAP tracker against alternative segmentation or tracking methods (e.g., Otsu thresholding, machine learning-based segmentation) to validate its robustness?

We have not done this. As it was our aim to create a broadly accessible method, we opted to use the features offered by TrackMate. In the future, we do hope to evaluate the Otsu/machine-learning based segmentation approaches, but it was beyond us for this revision.

To what extent do these LAP tracking parameters require optimization based on fluorophore brightness, molecular mobility, or frame rate?

They do. We have optimized the tracking parameters for both fluorophore brightness and molecular mobility, based on the binding of 4 different glycans to the CHO-MR/CHO cell line pair (please see response to earlier comment for a discussion regarding the results of this sensitivity analysis). These four glycans capture a binding range and therefore reflect varying molecular mobility and brightness. We have completely rewritten the section of the manuscript that describes this analysis to more clearly reflect our choices in this manner. In addition, we have expanded the discussion to more clearly describe these limitations. Regarding, the frame rate,

we have kept it the same as in our previous study on CHO-MR to remain consistent with this work (<https://pubmed.ncbi.nlm.nih.gov/34764473/>).

Was any experimental or manual validation performed to ensure tracking accuracy under the specific conditions used in this study?

For the CHO-MR cells, we have correlated all our data to data obtained by our manual Glyco-PAINT method. In addition, we also used biological validation of the tracking parameters. Firstly, we used the cell pairs of CHO-MR and CHO (that does not have the MR) as a positive and negative control. This allowed us to strictly isolate MR-only binding events. We defined the range of acceptable settings based as those in which our negative controls were negative and even our weakest binding positive controls were positive. On the DCs, we have added additional experiments (Figure 4 and 5) using dendritic cells and macrophages of a MR-knockout mouse. This allowed us to also look at MR-specific binding within the context of the other lectins present on DCs. These controls allowed us to experimentally validate the parameter choices.

Spot detection is critical in the Glyco-PAINT-APP study, as accurate identification of glycan binding events directly impacts tracking reliability and the resulting kinetic measurements. Did the authors manually verify a subset of detected single-molecule events and compare them to the automated output?

Yes. We have extensively compared the manual (Figure S2) output to the automated output for CHO-MR (Figure S4) and have confirmed that this is in line with the original study. As mentioned in the text, the manual version of the method is not effective for primary cells due to the introduction of averaging artefacts.

Additionally, were small changes in detection threshold tested for their effect on event counts or kinetics? The authors mention using thresholds of 5, 10, and 15, but a direct comparison of the resulting datasets is not provided. Since threshold selection can introduce detection bias - either by excluding weak but valid events or by including noise - it is important to explicitly assess the robustness of the analysis across different threshold settings and address potential bias in spot detection.

*Yes, they were. We have now included a spot detection threshold sweep in Figure S5 and S6 as part of the parameter sensitivity analysis for the CHO/CHO-MR dataset. From this analysis we concluded that thresholds above 4 can reliably discriminate MR-mediated binding events from background and therefore we concluded that any value above 4 would not result in detection biases arising from inclusion of background noise as suggested by the reviewer. Off-rates, on-rates and diffusion coefficients were independent of the threshold parameters. A detailed description and interpretation of the parameter sensitivity analysis is also added in lines under the subheading **Glyco-PAINT-APP** in the revised manuscript main text.*

3. Regarding the figure 2: What kind of “select squares criteria”—free or strict—was chosen in b and c? After watching the videos, the images confuse me a bit. It seems like in figure 2b, the square areas are quite noisy with high background in comparison to 2f. Was the same background correction applied to both images, or were different noise reduction methods used?

For both the indicated images, select squares criteria was set to “free”. We thank the reviewer for raising this point as the squares selection criterion was not mentioned in the methods section. We have now also included this in the experimental method detail of the revised manuscript.

Additionally, the squares represented in g and h do not correspond to the squares in f; rather, they are additional squares in different positions. Why were different squares analyzed instead of using the same ones as in f? Wouldn't it be more consistent to show and analyze the same squares (in f, g and h) as was done for the Trimannose Cluster (13) analysis? If different squares were intentionally chosen in f, g and h, what was the rationale for this selection? Given that the square selection method can significantly influence the interpretation of spatial binding behavior, it is important to clarify the selection criteria, ensure consistent background processing, and maintain comparability across panels.

Thank you for pointing out this mistake in the squares selection criteria for generating these images. We have reevaluated the squares selection criteria for all figures and revised the figures.

4. Regarding figure 4 and figure S7, where the binding parameters of glycosylated SLPs to cells were tested: The authors stated (lines 277-280) that the on-rates, off-rates, and diffusion coefficients were significantly increased for all glycosylated probes (1-6) but not for the non-glycosylated control (7). However, this does not fully align with the data presented in Figure S7.

Specifically, in the CHO-MR treated condition (figure S7, g and h) and the BMDC-treated condition (figure S7, a and c), on-rates and off-rates do not consistently increase - in some cases, they partially decrease, contradicting the claim of a uniform increase. Additionally, the on-rate for the non-glycosylated control antigen 7 also increases in certain conditions (figure S7, c, d, and h), which is inconsistent with the statement that only glycosylated probes exhibit this effect.

*Thank you for pointing at this inconsistency in the textual description of the data. In the current version of the manuscript we have revised this analysis and corrected data filtering mistakes in the original figures as mentioned by the reviewer. We have revised the text belonging to this statement under subheading **Uptake and antigen presentation of SLP glycoforms** and added additional figures to display the precise effects of CytD on kinetic parameters which are shown in Figure S14, and also in Figure S9 and Table S1 for the numeric values.*

Please comment on the discrepancy between the reported binding kinetics in figures 4 and S7 and the expected trends based on previous studies. Studies on mannose receptor (MR) and DC-SIGN binding kinetics (Feinberg et al., 2001; Taylor et al., 1992) suggest that trimannosylated ligands should exhibit lower off-rates than monomannosylated ones due to multivalent binding effects. However, figure 4 shows no such difference. Additionally, prior work on actin-dependent receptor dynamics (Lau et al., 2007) indicates that Cytochalasin D should disrupt clustering and reduce binding stability, rather than uniformly increasing on- and off-rates. Can the authors clarify how their findings align with these previously reported trends?

In their 2001 paper, Feinberg et al. ([https://www.jbc.org/article/S0021-9258\(21\)00140-X/fulltext#fig3](https://www.jbc.org/article/S0021-9258(21)00140-X/fulltext#fig3)), show the binding of a CfG-array to CTL-Domain-4 of the MR in steady state. What confuses me is two aspects of this study, namely that this is a steady state binding study with no off rates given, and that mannose still binds relatively well compared to some other high mannose structures (e.g. 264, 499, and 601). An additional factor that makes these data hard to compare is because their lectin reagent is a streptavidin tetramer of a single lectin domain, suggesting that the multivalent binding effect could be the result of the lectin reagent. The 1992 paper by Taylor and co-workers, again does not give off rates, but steady state binding of mannosylated-BSA, invertase and mannan to various binding domains. That aside, we too had expected to see a lower off rate for the trimannoside compared to a monomannoside and expected a lengthening of these off rates with CytD treatment. Whereas we did not observe significant differences in off rate between the mono and the trimannoside, we did see lowering of the off rate going from 1 to 2 to 6 copies of each of these sugars, suggesting a multivalent effect on a slightly larger distance scale. The fact that we did not see a uniform lowering of the off rate (as a proxy for reduced binding stability) is in line with our observations of our previous paper where we showed that the MR did not bind our ligand library in cluster form, instead it was single MR-ligand interactions responsible for binding. This combined with the lack of an cytosolic actin anchor of the MR and the off targets had us ascribe these observations to those parameters. However, to further clarify some of these issues (particularly on the DCs that also have other lectins), we have included the same set of experiments and correlation with MR-/- BMDCs (Figure 4/5) rather than the CytD experiments, which have now been moved to the SI.

5. The manuscript uses different Density_Ratio thresholds (>10 in case of macrophages and >2 for BMDCs). Can the authors clarify why different thresholds were chosen and whether these values were optimized based on a specific criterion? How does varying the Density_Ratio threshold affect the interpretation of the results? A direct comparison of results using different Density_Ratio values would help assess the robustness of the findings and eliminate bias in data interpretation.

There was no specific optimization criterion behind the use of different Density_Ratio thresholds; the choice of >10 for macrophages arose from the observation that BMDMs exhibit substantially higher levels of ligand binding events (by virtue of their higher MR-expression), which is also apparent in the absolute density values. To address this, the same figure with filtering at Density Ratio 2 is now included in the manuscript. As found for the optimisations described earlier, values do not deviate significantly upon this change in Density_Ratio. In the revised version of the manuscript we describe that not Density_Ratio but R_Squared is limiting parameter for squares selection in BMDM recordings. A supplementary figure is added that visually displays the discrepancy in square selection for three scenario's, R_Squared > 0.1, R_Squared > 0.9 and Density_Ratio > 10 (Figure 14A, C). Here it can be seen that typically for BMDM long duration tracks do not meet the criteria for good fit and Tau derivatization. Therefore it may be considered to reduce the selection criteria here and then for example look at different parameters that are not dependent on fitting but do reflect track lengths. We have thus picked Total Track Duration as the parameter to display, as it meets this requirement.

6. Regarding figure 5: In figure 5b, the square areas appear quite noisy with a high background compared to Figures 5a and 5c. This raises concerns about the high Density_Ratio threshold used for this dataset. Would it be possible that the polarization towards M1 macrophages

primarily detects strong clustering or receptor hotspots, rather than capturing the full range of receptor behavior? Would broader changes in receptor behavior, including moderate enrichment, be more effectively detected using Density_Ratio >2 instead of >10? A Density_Ratio >2 might be more suitable for analyzing broader spatial patterns, allowing for more variation in binding behavior to be captured, rather than focusing exclusively on the most intense binding regions.

Can the authors comment on this? I may be misinterpreting the reasoning behind the choice of Density_Ratio thresholds, so clarification on whether different thresholds influence the biological conclusions would be valuable.

As explained in previous comment, it is not so much the Density_Ratio but mostly the R_Squared that results in this square selection. For clarity additional tracking overlay figures with Density_Ratio 10 (instead of 2) have been added to Figure S13A.

7. Additionally, the figure 5 provides interesting insights into how macrophage polarization affects MR ligand binding, particularly with hexatrimannoside. However, testing only one ligand does not fully establish whether polarization alters MR binding specificity. Would testing additional glycan ligands (e.g., mono- and trimannosylated ligands) help determine whether MR binding preferences shift in M2 macrophages?

Thank you for this helpful suggestion. In response, we have now performed additional experiments including glycans 1 Mono, 6 Mono, 1 Tri, and 6 Tri—matching the probe selection used in earlier parts of the manuscript. This expanded dataset provides a more comprehensive view of how macrophage polarization influences glycan binding. Figure 5 has been revised accordingly to reflect these new findings.

Additionally, could receptor clustering contribute to prolonged dwell times in M2 macrophages rather than just increased MR expression? A fluorescence-based receptor clustering analysis could clarify whether M2 macrophages exhibit larger receptor aggregates, leading to longer binding events.

Thank you for this helpful suggestion. In both our earlier paper and this work, we have performed K-means clustering analysis based on the average x,y positions of the detected tracks for the macrophage experiments across all four glycan probes and the three adjuvants. From this analysis we could not detect a significant difference in clustering of the binding events for the polarization states of most of the tested ligands. Except for 1 Mono (8), which appeared to have a slightly higher clustering score for the M1 phenotype. This observation is somewhat analogous to what was observed for CHO-MR in our earlier work. These data support the hypothesis that – for soluble ligands at least – the MR does not require clustering for binding. See Figure S13C for the results of the clustering analysis.

The authors also state that binding events in the M2 state exhibited longer diffusion coefficients compared to the M1 or M0 state. However, from the data, the log diffusion coefficient appears nearly the same between M1 and M2 macrophages, with only the frequency of events increasing. This suggests that binding kinetics are altered by the number

of MR-positive events rather than by an inherent difference in receptor diffusion behavior. Can the authors clarify this point?

We have removed this statement.

Furthermore, in Figure 5B, it is noticeable that some tracks exhibit long track lengths, but these tracks are unfortunately not covered by the grids used in the analysis. Could this influence the interpretation of the results? If long tracks are systematically excluded from the analysis, the impact of extended receptor-ligand interactions may be underestimated. Would an alternative grid size or adaptive segmentation approach help capture these events more effectively?

*We indeed found that some tracks fall outside the selected grid regions, particularly in macrophage images where extended binding duration events are more commonly found. Our intention was to use an automated, grid-based segmentation approach, which applies stringent selection criteria to distinguish true signal from background noise. We have however in response to comment 5 of this reviewer reevaluated the square selection criteria for macrophages specifically and found that by lowering the $R_Squared$ criteria to 0.1 there tracks are more efficiently captured. This is also illustrated in Figure S15A, C and described under subheading **MR binding on macrophages***

Lastly, since M1 macrophages exhibit increased receptor shuttling, would comparing MR internalization rates clarify whether ligand binding is affected by receptor turnover rather than diffusion alone?

Thank you for this insightful suggestion. To address the potential role of receptor turnover in ligand binding dynamics, we analyzed the uptake of the glycan probes by macrophages. We do however find that uptake of the glycans is merely identical for the various polarization states so have updated this in the manuscript (Figure S12B).

8. For a balanced interpretation of the data, the discussion section should more transparently address study limitations. In particular, potential confounding variables such as receptor trafficking, glycan degradation, and pH effects need to be considered, as they could influence the observed binding kinetics. Additionally, the authors should explicitly acknowledge that correlation does not automatically imply causation, and alternative explanations for the reported effects should be discussed.

We have significantly altered the discussion to more transparently show the limitation of the method and of the analysis and interpretation of the data on the points suggested.

Reviewer #2 (Remarks on code availability):

The GlycoPaint-Pipeline repository provides a structured and well-documented analysis pipeline for single-molecule glycan-receptor interaction data. The authors supply detailed

installation instructions through a dedicated Installation.md file, guiding users through the setup of Fiji, TrackMate, Python, and R environments. The codebase includes both Python and R scripts and is organized with clear folder structures and metadata. Importantly, the repository also includes test datasets (Demo/ folder), such as All Tracks.csv and Experiment Info.csv, which allow users to verify installation and explore example outputs. This is a valuable addition that enhances the reproducibility of the pipeline. That said, I found the large number of files and subfolders a bit overwhelming to navigate, and it's possible I missed some details despite reviewing the materials thoroughly. While I'm not a programming expert, I do have more experience than a typical user, and from that perspective, the pipeline is moderately accessible. However, the overall usability would improve with better integration between the data, code, and documentation to make it easier to understand how the components relate and how to get started. To support broader use and reproducibility, the authors are encouraged to:

- Add a walkthrough or script that demonstrates how to use the demo dataset to generate key outputs or figures,

- Clearly link specific scripts to corresponding analyses or figures in the manuscript,
- Providing a requirements.txt or environment file to streamline setup.

Overall, the pipeline is a valuable and technically well-executed resource for specialized users and offers a strong foundation for reproducible glycan-binding analysis.

We thank the reviewer for critically assessing the code associated with our manuscript. Since reusability of the software is, besides sharing the biological data, one of the main points of the manuscript, we have in response to these comments revised the software manual and included more walkthrough scripts and also scripts that correspond to figures in the manuscript itself to further clarify the software. We had non-experts (i.e., undergraduate students) use and run the software to ensure its useability, but we gladly accept any further suggestions for improvement.

Reviewer #3 (Remarks to the Author):

Thank you for your time taken to review our manuscript.

Reviewer #4 (Remarks to the Author):

The authors describe a very interesting development of the Glyco-paint method to discriminate in a space resolved manner different kinetics of binding of glycans to myeloid cells. The pipeline analysis allowing glycans analysis is of high potential impact for the study of cells-glycan interactions. Nevertheless, there is a lack of evidence to suggest that the

binding observed is on Mannose Receptor and not a mixture of different C-Type lectin receptors on the cells. Similarly, the uptake of glycoclusters is not sufficiently supported.

We thank the reviewer for assessing our manuscript and acknowledging the potential high impact of the work. We will address the comments made by the reviewer regarding lectin specificity and uptake in detail in the below paragraphs.

Main comments

The authors mention the specific Mannose receptor in their study but other CLRs are able to recognize mannoses (like DC-SIGN for example), especially clustered mannose that might induce a lot of binding to other lectins. Is it possible to confirm that the binding observed is only on MR through labelling of endogenous MR (with antibody?) or MR knockout? Otherwise, the manuscript should be altered to reflect that the lectin binders are unknown.

We fully agree that multiple C-type lectins, including the murine homologues of DC-SIGN can also bind these clusters. To allow the separation of the contribution of the binding of these lectins to the overall biology, we have repeated all experiments with MR-/- BMDCs and macrophages to allow the specific identification of MR-specific binding events not seen in the knockout. These experiments show that the MR-binding events are essential for the uptake of the probes (as analysed by flow cytometry, but also by microscopy in Figure S11A, B). To our surprise, but fitting earlier data, there is some correlation with MR-binding and cross-presentation, but it is much weaker than we hypothesized. This offers a numerical confirmation of earlier literature discussions on this complex phenomenon.

Can the heterogeneity of binding observed be ascribed to the binding to different lectins?
Yes, but pinpointing exactly which lectin (other than the MR), or combination of lectins, is responsible for which set of binding events is beyond us at this point in time.

The experiments on BMDC and CHO were performed with different concentrations of glycan probes, respectively 5 and 1nM. Could you comment this difference?

These concentrations have been selected to obtain a sufficient number of binding events per recording. The BMDCs have lower levels of MR than the CHO overexpression system used. This required a higher ligand concentration to obtain a suitable number of events as this parameter is dependent on both receptor and ligand concentration.

The uptake of the SLP-glycomolecules is measured by flow cytometry, which could also show binding to the cell surface. It would be interesting to show by confocal microscopy that the SLP-molecules are located in vesicles inside the cells to confirm the uptake.

We have confirmed vesicular intracellular uptake for two of glycan SLPs using confocal imaging. This data can be found in Figure S11A, B.

Minor comments

Page 5 lines 140-147. Please explicit why the early observations on BMDC are surprising in more detail as it is the basis of the study and will help the reader to understand the questions asked.

We have rewritten this section of the text to make the weirdness of our first BMDCs observations more explicit.

Could you also explicit which probes were used here and the rationale behind this probe choice.

We chose the probes to cover the extremes and middle of the binding ranges to the MR observed on the CHO-MR cells.

Figure 3 showing the structures of the fluorescent sugars is introduced only later.

We also struggled with this, as our chemist's instinct told us that these figure should mentioned beforehand. However, for the order of the narrative, we wanted to mention the controversy that started this paper first, before going into details. To somewhat solve this, a specific reference to Figure 3 is still added at the correct point in the main text. As we have also introduced these probes before in Riera et al. 2021., we also refer to them via this reference in the text. If this is insufficient, please let us know and we will add a supplementary figure in the correct chronological position. Otherwise, we hope this suffices.

Figure 1 c) data inspection and curation figure is too small to be informative. The same applies to the schemes of resting and polarised cells that is maybe a bit oversimplistic.

In the revised version of the manuscript the figures are added with substantial higher resolution to allow zooming in to observe more detail. The illustrations of resting and polarized cells is replaced with a schematic drawing of the glycans also in response to previous comment.

It would be helpful for the reader to introduce the structure of the glycan probes and parts of the averaged data to show the rationale behind the study (fig S1 E-F).

We have added a clear reference to the structures of the glycan probes in the first paragraphs of the text, referring to the correct figure. We wish to highlight that these structures have been reported before by Riera et al. We also are showing the averaged data and resulting averaging errors in Figures 1 and S1.

It is surprising on Fig S1 F that the difference in koff between glycans 1 and 6 is not judged significant.

We agree that this is surprising at first sight. We did verify the analysis underlying this figure repeatedly but came to the same conclusion. To help understand this fact we have added the p-value to the plot instead of 'ns'. Additionally, we wish to emphasize that the data in Figure S1F is based on identical data as Figure 2J

Fig 3 b,c please move labels (eg 1tri) so they are visible.
Corrected.

Reviewer #5 (Remarks to the Author):

In the manuscript titled “Quantification of single molecule glycan-mannose receptor binding kinetics on myeloid cells reveals high subcellular binding heterogeneity” by Kas Steuten, Johannes J.A. Bakker, Ward Doelman, Diana Torres-García, Roger Riera, Lorenzo Albertazzi and Sander I. van Kasteren, the authors describe an automated image processing pipeline they refer to as Glyco-Paint-APP that aims to extract sub-cellular glycan interaction kinetics. The work describes a spatially resolved quantitative image analysis approach that is applied to a method described in a previous study by Riera et al. (2021) Nature Chem. Biol. titled “Single-Molecule Imaging of Glycan–Lectin Interactions on Cells with Glyco-PAINT”. The application of their pipeline allows them to quantify carbohydrate lectin binding kinetics based with single molecule resolution.

While I am not an expert in the field of myeloid cells or lectin binding, I find the method description for quantitative spatially resolved kinetic measure of glycan interaction with receptors from glyco-PAINT experiment to be interesting and of wider relevance. Steuten et al. demonstrate using the image-analysis pipeline developed in FIJI with interface to TrackMate (Glyco-PAINT-APP), that receptor kinetics are indeed spatially inhomogeneous. Thus the technical development and the finding both form novel aspects of their work, improving on previous approaches. Using glycosylated synthetic long peptides (SLPs) they quantify immune cell uptake of SLPs and the role of actin polymerization. They find the residence time of glycan-SLPs to be the best positive predictor of cross-presentation in dendritic cell co-culture with CD8 T-cells. They also find the on-rates and diffusion coefficients in carbohydrate binding of mannose receptors (MR) correlate with polarization state of macrophages. Taken together these results suggest a utility for sub-cellular analysis using the tools developed for immune-cell dynamics and the role of carbohydrate ligand-binding kinetics.

We thank the reviewer for careful reading of the manuscript and positive assessment.

1) Figure 1c: Please add units to the diffusion coefficient and track duration plots. Unless they are normalized, in which case the legend needs to clarify this.

We have purposely not put the units in the subfigures of Figure 1 because they are meant as representative data to showcase the technique and do not necessarily have to be interpreted with units. If this is deemed fundamentally incorrect, we can however include the standard units as we did further in the paper, if the reviewer would so like.

2) Figure 1 plots appear to be screenshots of the outputs and the software interface and could be improved upon with high resolution figures.

In the revised version of the manuscript figures have been included as high-resolution files that maintain text and image quality.

3) Multiple figure legends have font sizes that are quite variable making them at times hard to read (e.g. in Fig. 4).

This has been addressed in the current version.

4) The brightfield images in Fig. 5 lack contrast. It might help to outline the features that the authors wish to highlight by outlining cells, similar to what has been done by them in Fig. 1.

We thank the reviewer for this comment and have addressed the resolution and image quality.

5) I found the instructions run and install the script somewhat confusing beginning with system configuration, paths and getting the FIJI plugin to work. The authors could clarify the installation procedure for usability of their code. I have not reviewed the code line-by-line since I was unable to actually run it.

In response to comments by this and other reviewers the software manual and instruction have been heavily revised and proofread by users that er not related to the project.